# Tree ring isotopes reveal an intensification of the hydrological cycle in the Amazon

Bruno B. L. Cintra [1,2,7] ✉, Emanuel Gloor [2,7], Jessica C. A. Baker [3,7], Arnoud Boom [4],
Jochen Schöngart [5], Santiago Clerici [2], Kanhu Pattnayak [6] & Roel J. W. Brienen [2,7]

Over recent decades the Amazon region has been exposed to large-scale land-use changes and global warming. How these changes affect Amazonia's hydrological cycle remains unclear as meteorological data are scarce. We use tree ring oxygen isotope records to confirm that the Amazon hydrological cycle has intensified since 1980. Diverging isotopic trends from terra firme and floodplain trees from distinct sites (approximately 1000 km apart) in Western Amazon indicate rainfall amounts increased during the wet season and decreased during the dry season at large-scale. Using the Rayleigh distillation model, we estimate that wet season rainfall increased by 15–22%, and dry season rainfall decreased by 8–13%. These diverging trends provide evidence, independent from existing climate records, that the seasonality of the hydrological cycle in the Amazon is increasing. Continuation of the observed trends will have a pervasive impact on Amazon forests and floodplain ecosystems, and strongly affect the livelihoods of the regional riverine communities.

There is growing concern about the future of the Amazon forests with several recent reports[1,2] suggesting the system could be close to a 'critical transition', interpreted as irreversible large-scale forest loss due to the interactive impacts of deforestation, heat and drought on local climate. Given Amazonia's importance for global biodiversity[3], terrestrial carbon stocks[4] and livelihoods[5] it is important to evaluate these concerns. Large-scale deforestation could cause reductions in precipitation[6] and land surface temperature increases[7] which are further affected by fossil fuel burning induced climate change[8] and by reductions in precipitation recycling due to vegetation responses to elevated atmospheric $CO_2$ concentrations[9,10]. Early estimates by General Circulation Models (GCMs) studying the combined effects of deforestation and climate indicated a severe drying of the Amazon, resulting in widespread dieback of the Amazon forests[11]. While more recent modelling predictions suggest such an extreme scenario may be unlikely, there is still considerable variation in rainfall projections for the Amazon varying from drying to wetting, and critical transitions could result[12].

Rainfall changes are thought to be a central component for potentially causing an irreversible transition of Amazonia's forests[2,6,11]. Obtaining more reliable data on changes in rainfall over recent time is thus a pre-requisite to understand in what direction the system is heading, and to determine which global climate model predictions are consistent with observations and thus

more likely to be realistic. Observational evidence indicates that the hydrological cycle of the Amazon has been changing in the past few decades with trends varying between regions and seasons. Some studies suggest that the south of the Amazon is getting drier[13–16], while others show wetting in large parts of the basin[15,17,18], with increased frequency of severe droughts and floods[19,20]. A reliable quantification of these changes is needed to understand contributions of anthropogenic drivers (i.e. deforestation and climate change) versus natural climate variability and to assess risks for the future[21]. However, these efforts are hampered by a lack of meteorological data. Rain gauge stations are sparsely distributed across the basin (only one station per 150,000 km²), their operation is often discontinuous, and methodologies are changing over time[22,23]. Remote sensing and data assimilation are becoming increasingly more accurate and reliable, but time series are relatively short and there is still disagreement between these products[18,23,24]. River level data may offer a more reliable, long-term record, but the varying seasonality of river levels of the many tributaries of the Amazon River may mask the timing and strength of the changes, especially for the dry season[25]. Thus, more information to evaluate the response of the Amazon basin to climate change is needed.

New information to aid our understanding of past variability of the hydrological cycle can be obtained from natural climate proxies. Different

[1]School of Geography, Earth and Environmental Sciences, University of Birmingham, Birmingham, UK. [2]School of Geography, University of Leeds, Leeds, UK. [3]School of Earth and Environment, University of Leeds, Leeds, UK. [4]School of Geography, Geology and the Environment, University of Leicester, Leicester, UK. [5]Ecology, Monitoring and Sustainable Use of Wetlands (MAUA), National Institute for Amazon Research, Manaus, Brazil. [6]Hadley Centre for Climate, Met Office, Exeter, UK. [7]These authors contributed equally: Bruno B. L. Cintra, Emanuel Gloor, Jessica C. A. Baker, Roel J. W. Brienen. ✉e-mail: b.ladvocat@bham.ac.uk

archives can be used for this purpose including ice cores[26], lake sediments[27], speleothems[28], and tree rings[29]. In forested regions, tree rings can provide climate proxies with annual resolution, but finding suitable tropical species that form reliable annual rings can be difficult. In addition, growth of tropical trees in moist and warm climates is generally not very sensitive to interannual climate variation[29]. An approach that partly overcomes these challenges is the analysis of stable oxygen isotopes in the cellulose of tree rings. This approach results in excellent signal strength, thus allowing more reliable reconstructions with fewer samples compared to standard ring-width chronologies[30,31]. Oxygen isotopes in tree rings ($\delta^{18}O_{TR}$) to a large degree reflect the $\delta^{18}O$ of source water (mostly rainfall)[32,33], which for the Amazon basin is controlled by the depletion of heavier isotopes in water vapor due to rainout of heavier water molecules ($H_2^{18}O$) during moisture transport over land (i.e. Rayleigh distillation of water in the atmosphere)[34–36]. Due to this mechanism, $\delta^{18}O_{TR}$ from western locations in the Amazon have been found to closely reflect the amount of rainfall over large regions of the Amazon basin and provide data representative of climate variability at large scale[30,31,37,38].

Annual ring formation in tropical trees is generally triggered by the annual cessation of cambial growth during a distinct dry season, restricting growth to the wet season[39]. An exception to this is trees from Amazon floodplains, where annually recurring multi-month floods create anoxic soil conditions, that impede cambial activity and restrict the growth of floodplain trees to the terrestrial (i.e. non-flooded) phase[40]. For many regions in the Amazon, the terrestrial phase of floodplain forests coincides with the local and Amazon-wide dry season (Supplementary Fig. 1). As a result, growth of floodplain and terra firme trees (i.e. non-flooded) are out of sync, occurring during the dry and wet seasons respectively (Fig. 1). These differences should allow for a seasonal reconstruction of the regional hydrological cycle at large-scale. Here, we pioneer this approach to reconstruct changes in rainfall seasonality from 1980 to 2010 by comparing tree ring oxygen isotope series ($\delta^{18}O_{TR}$) from a floodplain species (*Macrolobium acaciifolium*) and terra firme species (*Cedrela odorata*) from the western and south-western Amazon, with a climatic footprint that largely coincides with the Amazon-wide region and climate (Supplementary Fig. 2 and Supplementary Fig. 3).

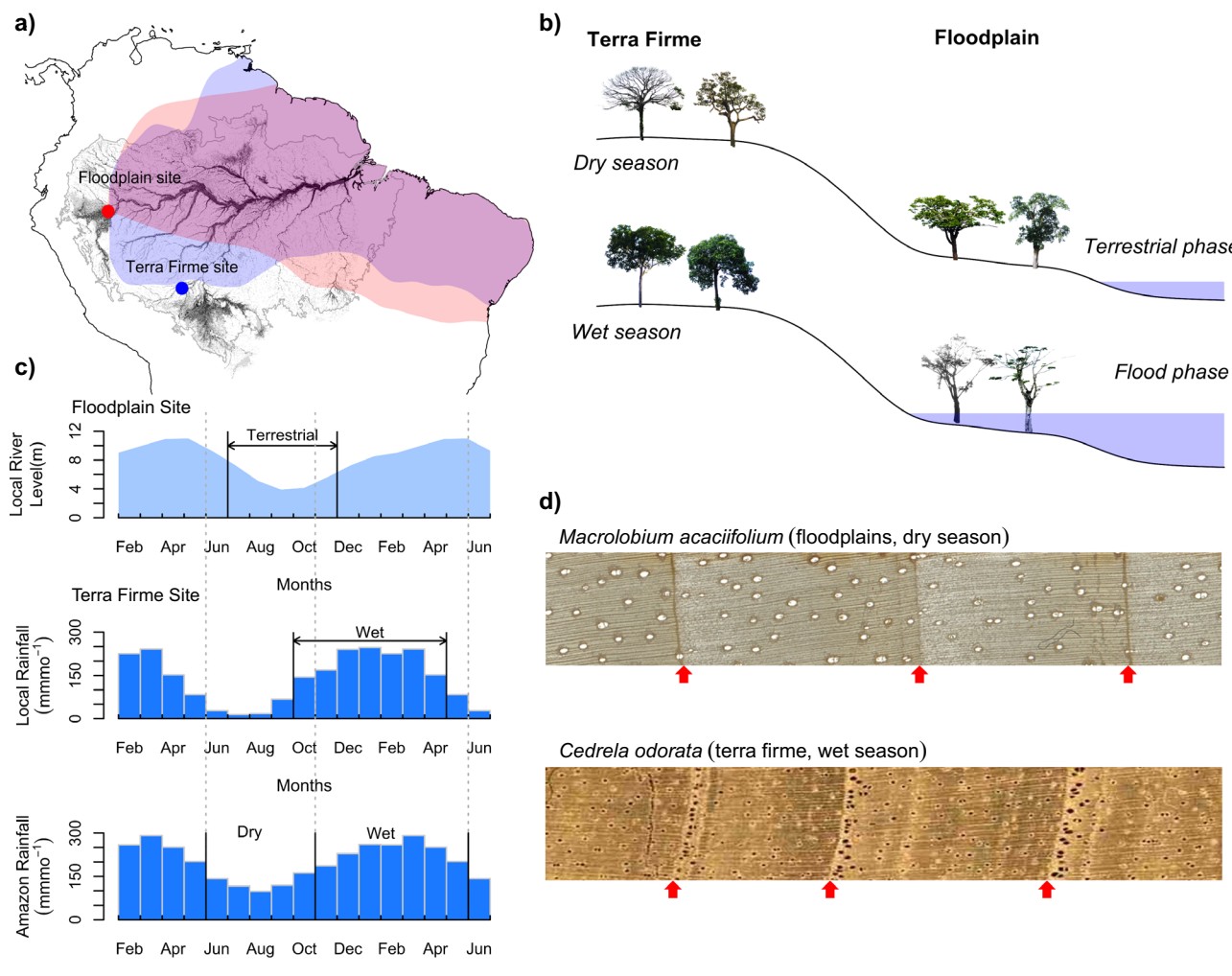

**Fig. 1 | Tree growing seasons at the seasonally flooded and the terra firme forests match the Amazon-wide dry and wet seasons, respectively. a** Map of the Amazon basin[97,101,102] showing the location of the floodplain site (red) and the terra firme site (blue). Black area corresponds to (seasonal) wetlands, rivers and seasonal floodplains. Color shading indicate the upwind regions of moisture transport from each site, derived from the HYSPLIT back trajectories (see Supplementary Fig 3 and Methods section). **b** Schematic diagram showing the seasonal phenological differences between terra firme and floodplain environments. **c** Seasonal variation of the local river level at the floodplain site, rainfall for the Amazon Basin, and local rainfall at the terra firme site. **d** Images of the annual growth rings of *Macrolobium acaciifolium* and *Cedrela odorata* tree species used in this study, with red arrows indicating ring boundaries. In (**c**), Arrows indicate the growing seasons of the trees, and vertical lines indicate the seasonal boundaries defined by rainfall and river levels. Vertical dotted lines extends the boundaries of the Amazon-wide seasons (bottom panel) for a visual comparison of how they overlap with the local growing seasons at each of the sites. Note that for calculations using rainfall data, we disregarded the shoulder months of those seasons, i.e. the months immediately next to the black vertical lines (details in Methods).

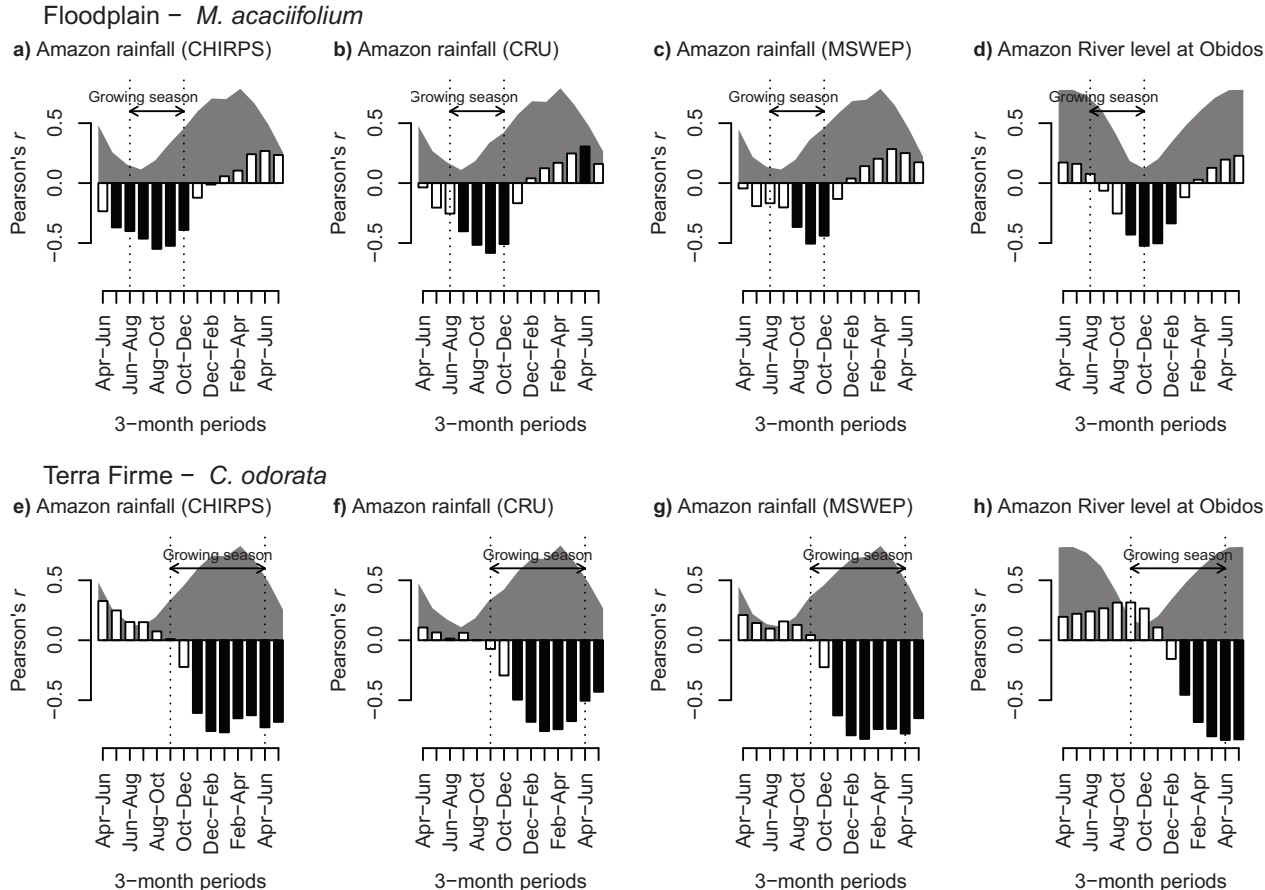

**Fig. 2 | δ¹⁸O_TR records are correlated with inter-annual variation of hydroclimatic variables in the Amazon Region. a-d** Correlations with the $\delta^{18}O_{TR}$ records of the floodplain trees. **e-h** Correlations with the $\delta^{18}O_{TR}$ records of the terra firme trees. Bars indicate the Pearson's $r$ between $\delta^{18}O_{TR}$ and 3-monthly means of rainfall or river levels. Filled bars indicate significant correlations with 95% confidence interval. Shaded areas show seasonal variation in precipitation in the Amazon region

**(a,b,c,e,f,g)** or Amazon river levels at Obidos **(d,h)**. Dotted lines and arrows indicate the growing season of the floodplain and terra firme trees. The x-axis spans the full hydrological year, and therefore extends across two calendar years. Months January to May in the three-month periods on the right-hand side of the x-axis correspond to the year after the onset of the ring formation.

## Results

### Seasonal precipitation signals recorded by floodplain and terra firme tree ring δ¹⁸O

The interannual variation of $\delta^{18}O_{TR}$ of both species is negatively related to the interannual variation in Amazon-wide precipitation during their respective growing seasons (Fig. 2). Peak rainfall correlations for the floodplain $\delta^{18}O_{TR}$ chronology occurred from August to November, within the growing season for *M. acaciifolium* which extends from July to November (Fig. 2a-c), while the terra firme $\delta^{18}O_{TR}$ chronology correlations peak from January to May, almost entirely within the growth season for *C. odorata* which extends from November to April (Fig. 2e–g). Correlations that extend outside the trees'growing seasons are generally weaker, which could be due to interannual variation in the growing season length of the trees.

Climate correlations for the record of *M. acaciifolium* are for a shorter seasonal window. This could reflect the short, non-flooded period during which this species mainly grows[40], or be due to differences in methodology as we only used the middle portion of the rings for this species to avoid interference of river water $\delta^{18}O$ and capture climate signals of the peak growing season[41]. Correlations for the *C. odorata* record cover a longer period coinciding with the longer growing season of this species[30]. The correlations for this record lag the onset of the species' growing season by one month probably due to a lag between leaf flush, the start of cambial activity and xylogenesis[42].

The two $\delta^{18}O_{TR}$ records also show close correlations with Amazon River level records measured at Obidos (Fig. 2d, h). The periods of significant correlations correspond very well with the months of low river levels for *M. acaciifolium*, and with the months of high river levels for *C. odorata*. Because of lag times between Amazon precipitation and river level[43], these correlations are ca. 3-5 months later than the rainfall-$\delta^{18}O_{TR}$ correlations. All correlations with climate were very similar when we averaged rainfall over the Amazon region or over the estimated regions of moisture transport (Supplementary Figs. 4 and 5). This large scale climatic footprint of the $\delta^{18}O_{TR}$ records is further evidenced by spatial correlation analyses of $\delta^{18}O_{TR}$ with gridded precipitation datasets (Supplementary Fig. 6).

### Estimates of rainfall changes inferred from trends in δ¹⁸O_TR

The two $\delta^{18}O_{TR}$ chronologies show diverging long-term patterns, with a positive trend ($p < 0.05$) in the *M. acaciifolium* record and a negative trend ($p < 0.1$) in the *C. odorata* since the mid 1970s. The divergence between the two records since mid 1970s is greater than at any other time in the period covered (Supplementary Fig. 7). The trend in the *M. acaciifolium* record starts a few years earlier than the trend in the *C. odorata* record. To estimate the magnitude of changes in precipitation, evapotranspiration, or moisture influx required to explain the trends in the $\delta^{18}O_{TR}$ records, we used a Rayleigh distillation model (Eq. 3, Fig. 3, see Methods for details). For these estimates we calculate changes for the period of 1980-2010, when there was an increase of 1.14‰ in the dry

season record for *M. acaciifolium* and a decrease of 0.90‰ in the wet season record of *C. odorata* (Fig. 3c).

In the simplest version of this model, we assumed that there were no changes in moisture influx into the basin, or evapotranspiration along air parcel paths, and thus attributed all observed changes in $\delta^{18}O_{TR}$ to changes in precipitation along the water vapour path. This results in an estimated increase in wet season precipitation of 5% per decade (i.e. $\Delta P/\Delta P_i$, 15.5% since 1980 C.E.) (Fig. 3e, Supplementary Fig 8b, Table 1) and an estimated decrease in dry season precipitation of 4.4% per decade (−13.5% from 1980–2010 C.E.) (Fig. 3d and Supplementary Fig 8a, Table 1).

These estimates however are sensitive to various assumptions. Firstly, this model version assumes no trend in the coastal (i.e., initial) air column water vapour content ($N_i$). However, observed sea surface temperatures have increased by approximately 0.5°C since 1980 C.E. over the tropical North Atlantic, which would result in an 4% increase in air water holding capacity according to the Clausius-Clapeyron equation for saturated vapour pressure (i.e., 7% for each °C[44]). Assuming the change in mass of water in atmospheric inflow is linearly proportional to the ratio of saturated vapor pressures in 2010 and in 1980, accounting for this increase in water vapour in the Rayleigh model (i.e., changing $N_f - N_i$, or $\Delta N$, see Eqs. 2, 3) results in larger estimates for increases in wet season precipitation (i.e. 5.9% instead of 5% per decade) and weaker drying trends for the dry season (i.e., −3.6% instead of −4.4% per decade, Case 1 in Table 1). Independent records from ERA5 indicate that wet season (Nov-March) water vapour inflow increased

by 8%, but show no support for dry season changes in vapour inflow (Supplementary Fig. 3ef). Using the changes from ERA5 would result in a larger difference between wet and dry season trends.

We further assumed there are no changes in the total evapotranspiration of the forest. In our estimates, we considered that evapotranspiration recycles 40% of total dry season rainfall and 32.5% of total wet season rainfall[45]. Deforestation or $CO_2$-induced stomatal closure may result in decreases in evapotranspiration[6,46], while warming induced vapour pressure deficit may increase evaporative demand[47]. These opposing effects may offset each other[48], which could explain why no changes in evapotranspiration are evident within a variety of dataset types and sources for the Amazon[49]. While the extent of shifts in evapotranspiration at large-scale in the Amazon remains uncertain[47], recent evidence suggests that Amazon evapotranspiration may have decreased by approximately 6% in the past 30 years[50]. The contribution of total evapotranspiration to rainfall can influence the degree of Rayleigh distillation effect during moisture transport (Fig. 3b) and thereby add uncertainty to our estimates (Eqs. 3 and 4). We conducted a sensitivity analysis to assess variations in our estimates under different scenarios of changes in evapotranspiration during the analysed period. This sensitivity analysis shows that our estimates of precipitation changes are not very sensitive to the effects of changes in evapotranspiration on the Rayleigh distillation: the Rayleigh model shows that a 1-1.5% change in evapotranspiration per decade has only a minor (±0.5 to 0.8% per

**Table 1 | Predicted rainfall changes are sensitive to sources of uncertainty in Rayleigh model parameters**

| Case | Dry season | | | Wet season | | |
|---|---|---|---|---|---|---|
| | Parameter change from 1980-2010 | Predicted rainfall change (%) | | Parameter change from 1980-2010 | Predicted rainfall change (%) | |
| | | Per decade | Total | | Per decade | Total |
| **Baseline** | | | | | | |
| **0.** No change in Rayleigh model parameters from 1980 to 2010[a] | - | −4.4% | −13.5% | - | +5.0% | +15.5% |
| **Tropical Atlantic Sea Surface Temperature change from 1980 to 2010 causing increase of** | | | | | | |
| **1.** water vapor in air entering the basin caused by an increase in evaporation from the sea[b] | +4% | −3.6% | −11.2% | +4% | +5.9% | +18.3% |
| **2.** $\delta^{18}O$ of moisture entering the basin due to decrease in sea-air fractionation[c] | +0.04‰ | −4.2% | −13.0% | +0.04‰ | +5.2% | +16.1% |
| **Upper atmosphere temperature change causing increase of** | | | | | | |
| **3.** fractionation temperature | +1°C | −4.2% | −13.1% | +1°C | +5.2% | +16.1% |
| **4.** combined effect of 1, 2 and 3. | - | −3.3% | −10.3% | - | +6.3% | +19.6% |
| **5.** Combined effect of Case 4 and change in vegetation evapotranspiration from 1980-2010[d] | −5% | −3.9% | −12.0% | −5% | +5.6% | +17.2% |
| | +5% | −2.8% | −8.6% | +5% | +7.1% | +22.0% |
| **Sensitivity of Case 4 to** | | | | | | |
| **6.** cloud condensation temperature | −5°C | −3.4% | −10.4% | −5°C | +6.4% | +19.7% |
| | +5°C | −3.3% | −10.2% | +5°C | +6.3% | +19.4% |
| **7.** continental gradient ($\Delta\delta^{18}O_{cont}$)[e] of $\delta^{18}O$ in precipitation | −0.27‰ | −1.9% | −5.8% | −0.36‰ | +5.0% | +15.5% |
| **8.** recycling ratio, $r_{E:P}$ in 2010[f] | −10% | −3.5% | −10.9% | −10% | +6.7% | +20.7% |
| | +10% | −3.1% | −9.7% | +10% | +6.0% | +18.4% |

Several model predictions are shown, each incorporating one or more changes to the baseline parameters (Case 0). The effects of these parameter changes are reflected in the differences between Cases 1–5 and Case 0, and between Cases 6–8 and Case 4. All predictions are reported both as percentage change per decade and as percentage change from 1980–2010, each relative to 1980 values.
[a]All Rayleigh model parameters kept constant between 1980 and 2010 as per best estimates, see Methods section – Eq. 3.
[b]Predictions assuming an increase in coastal air column water vapour ($N_i$) between 1980 and 2010 (i.e., $\Delta N$) due to increase of 0.5°C in Sea Surface Temperatures inferred from Had-ISST1[93].
[c]Predictions assuming a decrease in sea to air fractionation due to the 0.5°C increase in SST[90].
[d]Predicted precipitation change assuming 5% change in evapotranspiration between 1980 and 2010.
[e]Using difference between $\delta^{18}O_{coast}$ to $\delta^{18}O_{site}$ predicted by the isotope enabled Hadley Centre Climate model[92], instead of observations in precipitation at each of the sites and at Belém, Brazil from the GNIP database. See Supplementary Fig. 3a-d for geographical locations of modelled $\delta^{18}O$.
[f]$r_{E:P}$ is the ratio of evaporation to precipitation, i.e. precipitation recycling. Higher recycling ratios tend to weaken the Rayleigh distillation (e.g. Fig. 3b). We used $r_{E:P}$ present time values of 0.4 for the dry season and 0.325 for the wet season[45]. In all cases, asymmetric changes in evaporation and precipitation imply that past $r_{E:P}$ is not the same as in the present.

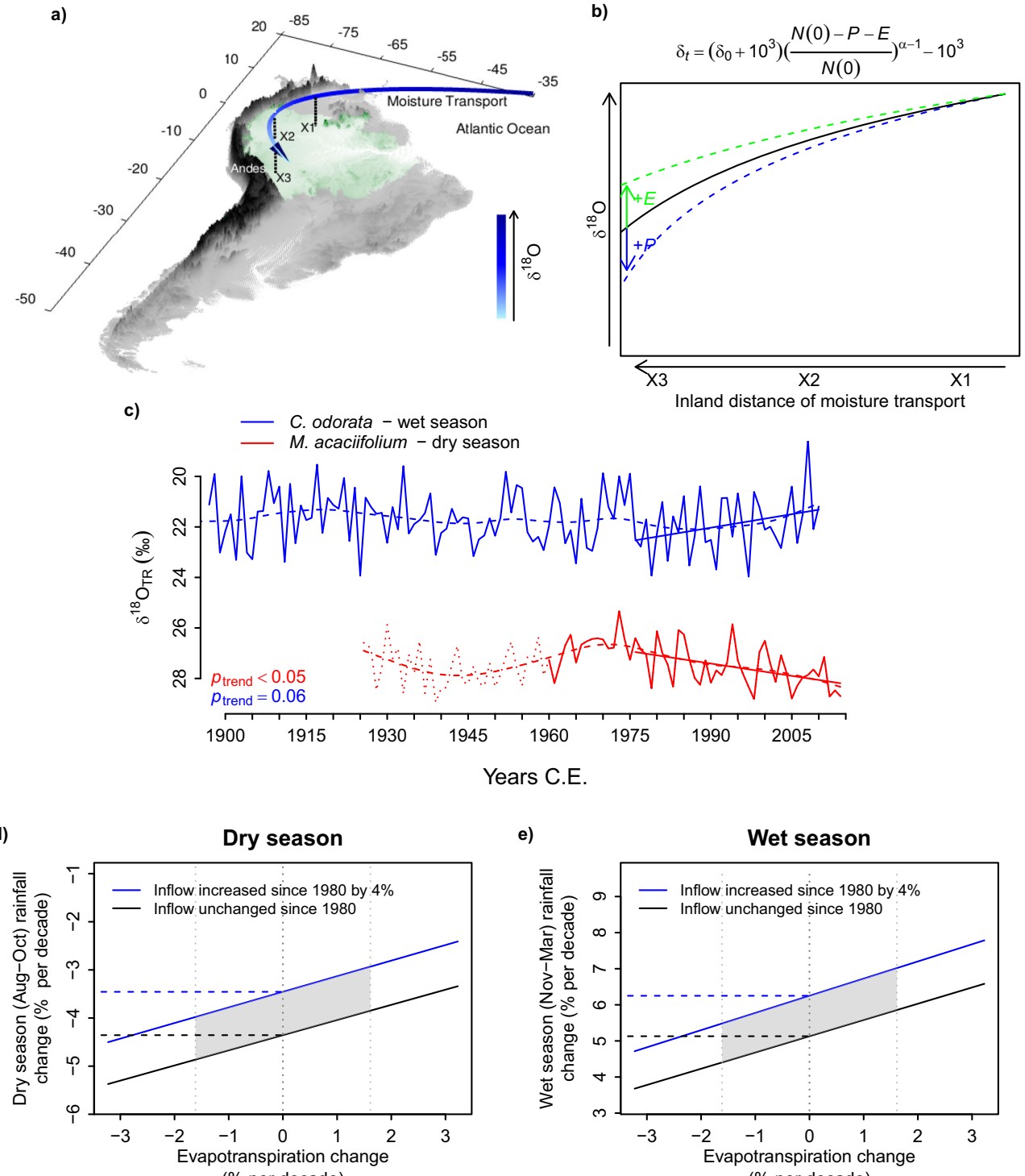

**Fig. 3 | Trends in the $\delta^{18}O_{TR}$ records reveal contrasting changes in the amount of rainfall during the dry and wet seasons. a** Diagram of the isotopic distillation of inland transport of atmospheric moisture over South America[103,104]. **b** Graphic representation of the changes in the $\delta^{18}O$ of atmospheric moisture based on a Rayleigh distillation model (shown on top of the graph, rearranged from Eq. [1]), with green and blue lines indicating the expected effects of increasing evapotranspiration (E) and precipitation (P), respectively. **c** $\delta^{18}O_{TR}$ chronologies, with blue lines showing the wet season record obtained from *C. odorata* trees from terra firme forests, and red line showing the dry season record obtained from *M.*

*acaciifolium* trees from floodplain forests. Straight lines indicate significant trends at 90% confidence interval. **d, e** Inferred changes in rainfall since 1980, calculated from a Rayleigh distillation framework based on the $\Delta\delta^{18}O_{TR}$ trends (see methods). Lighter vertical stippled lines indicate an evapotranspiration change scenario corresponding to 0% and ±5% over the entire analysed period, and shading indicates possible variations arising from uncertainties in estimates of evapotranspiration, moisture flux and vapor $\delta^{18}O$ (see also Table 1). Note that in (**c**), the $\delta^{18}O$ axis is inverted.

decade) effect on our estimates of precipitation changes for the two seasons (Fig. 3, Table 1, case 5). Our analysis further shows that these estimates are insensitive to assumed condensation temperature of water vapour in the clouds (Table 1, case 6).

Additional smaller uncertainties in our estimates arise from possible changes in the contribution of leaf water enrichment to $\delta^{18}O_{TR}$, evaporative enrichment of soil water, in the isotopic composition of water vapor entering the basin, in rainfall type or intensity, and from uncertainties related to the magnitude of the continental difference of $\delta^{18}O_{coast}$ to $\delta^{18}O_{site}$, and the assumed recycling ratio of evapotranspiration over precipitation, $r_{E:P}$. See Table 1 for all quantified uncertainties arising from these factors, and supplementary materials for a full discussion of how these factors may affect our estimates.

## Discussion

Our results show that floodplain and terra firme trees record the climate for different seasons at large-scale (Fig. 2). When analysed together these records provide more detailed, seasonally resolved climate reconstructions for the Amazon compared to using one single species. To the degree that $\delta^{18}O_{TR}$ accurately reflects rainfall $\delta^{18}O$, these trends are due to long-term changes in the rainout of heavy water ($H_2^{18}O$) along the air parcel trajectory from the tropical north Atlantic to the site of precipitation[34,35]. Previous analysis for *C. odorata* from the terra firme site has shown that the total cumulated precipitation over the air parcel path over land is a very strong predictor of inter-annual variation in $\delta^{18}O_{TR}$[31] giving confidence that the signal is governed by Rayleigh distillation during moisture transport over land. Analysis of atmospheric back trajectories of moisture transport for the two study sites show that they have similar paths that cover a large portion (60-70%) of the Amazon basin, with a climatic footprint that is highly representative of Amazon-wide rainfall (Fig. 1, Supplementary Figs. 2-5). This large-scale spatial footprint is further confirmed by spatial correlation analyses of these records with gridded precipitation datasets (Supplementary Fig. 6). The diverging long-term trends in $\delta^{18}O_{TR}$ are therefore interpreted to reflect diverging seasonal trends in precipitation for the same central region of the Amazon basin.

Taking identified uncertainties into account (see Table 1, Fig. 3, Supplementary Discussion), we estimate wet season rainfall to have increased by between 5 to 7.1% per decade (15.5-22% since 1980) and dry season rainfall to have decreased by 1.9 to 4.4% per decade (5.8-13.5% since 1980, Fig. 3c). This provides evidence for increases in precipitation seasonality using a record that is independent from current climate records, and that integrates variability in rainfall over a large area within the Amazon basin. Our analysis is based on a mechanistic understanding of the association between climate and $\delta^{18}O_{TR}$ and overcomes some of the limitations of spatial and temporal scarcity of meteorological data in the Amazon[22].

To what degree are tree ring estimates of Amazon precipitation changes consistent with observations? To evaluate this question, we compared our $\delta^{18}O_{TR}$ to rainfall and river stage records for the two seasons (Fig. 4). Interannual variation and long-term trends in $\delta^{18}O_{TR}$ are well correlated with variation in precipitation and with river level (see Fig. 4, please note the inverted scale for oxygen isotopes). Our estimates from tree ring oxygen isotopes suggest rainfall reductions in the dry season ($-1.9$ to $-4.4\%$ per decade) are largely within the range of those derived from the different gridded precipitation datasets used here (CRU: $-0.3\%$, CHIRPS: $-6.3\%$, MSWEP: $-1.6\%$ per decade), but imply stronger rainfall increases during the wet season ($+5$ to $+7.1\%$ per decade) than estimated from all three datasets (CRU: $+1.6\%$, CHIRPS: $+1.6\%$ and MSWEP: $+0.3\%$, per decade). This discrepancy in the estimates of rainfall changes may possibly be caused by differences between the footprint of the tree ring isotope series (i.e. the region of moisture transport from coast to sampling sites, Fig. 1, Supplementary Figs. 2 and 3) and the spatial distribution of the stations that these climate records relied on. The difference in the estimates from different gridded datasets also highlight the uncertainties arising from their use to assess long-term climate trends in the Amazon region. Despite this difference, our estimates of rainfall changes largely agree with the few

published estimates of precipitation changes in Amazonia, which consider different datasets. Few studies report similar magnitudes to what we found here, with a reduction of up to 4.4% per decade in the dry season and an increase of up to 10% per decade in the wet season depending on region[17,19,51], and one recent study (based on observations and reanalyses) reports an increase of 6% per decade in rainfall seasonality, given by the difference between maximum and minimum rainfall amounts per year, since 1979[52]. Diverging seasonal trends have also been detected regionally, with up to ~5% (per decade) dry season drying in the south and similar wet season wetting in the north portion of the basin[53,54]. Interestingly, these results seem to go in the same direction of ours, even though the spatial footprint of our records does not capture all of the southern Amazon during the dry season. Here we cannot make any inference regarding spatial heterogeneity of the trends, but it is possible that to some extent our integrated measure of the regional hydrological cycle reflects spatially heterogeneous climate changes in the region.

In all, our tree ring trends provide strong support for a substantial increase in the seasonality of Amazon rainfall, with the wet season becoming wetter and the dry season drier. We find no support for previous GCM predictions for a general drying of the Amazon basin[11,55–57], or for a general wetting[58]. Instead, our results point to an increased seasonal cycle, as observed before in both precipitation and river level records[13,17,19,52,53,59]. These latter authors have analysed possible drivers for these trends, and find changes are related to differences between the tropical north Atlantic and Pacific SST anomalies, in part associated with natural variability, but a more rapid warming in the Atlantic is likely to continue due to anthropogenic forcing[59]. These large-scale changes have led to an increase in the upwelling of the Walker cell branch over Amazonia[59,60].

Existing tree ring records from the Amazon provide some support for the increase in wet season precipitation suggested by our records. An extended $\delta^{18}O_{TR}$ record (1980 to 2020) from *Cedrela odorata* located in the nearby Brazilian southwest Amazon[61] shows a similar upward trend in precipitation up to 2020, despite the strong 2015 El Niño (i.e. dry anomaly). Tree ring-widths from south-west Amazon-Andes also capture this wetting trend locally[62], and one study from eastern Amazon points out that climate signals in the tree rings became weaker after 1990's[63], which can be an indication that growth limitation by rainfall is decreasing, even in the absence of trends. Other ring-width studies from different parts of the Amazon don't seem to capture this change[64–67]. The reasons why tree ring-width do not capture these changes may vary. Firstly, trees growing in humid conditions during the wet season may not be limited by the precipitation ammounts, and thus don't benefit from the increase in rainfall. Secondly, tree ring studies often focus on the high-frequency variability, and may not reproduce multidecadal trends. In addition, detecting trends in ring-width series requires consideration of several ontogeny and other biases due to demography or field sampling strategies, which are often not considered[39,68,69]. In comparison to tree ring widths, $\delta^{18}O_{TR}$ records are superior proxies for hydroclimate as the signal in these two sites arises predominantly from variation in precipitation $\delta^{18}O$ and is thus relatively independent of plant physiological effects[30,41] and do not require detrending to correct for ontogenetic effects.

Though the climate changes inferred from the $\delta^{18}O_{TR}$ chronologies may seem large, the severity of recent climate extremes is consistent with expected consequences of climate change[21], and trends have persisted over time. Our records only extended to 2010 and 2014, but extreme floods of the Amazon River have continued to happen with high frequency. The four highest flood-season river levels (2021, 2012, 2009 and 2022) and three lowest dry-season river levels (2024, 2023, 2010) all occurred during the last 15 years[70]. Of those, the most extreme levels for the entire 122-year record at the port of Manaus were recorded in the past 3 years, indicating that the intensification of the hydrological cycle is persistent, with increased rainfall seasonality leading to more frequent droughts and floods[19,20]. As a result, the maximum flood area of the Amazon River has extended by 26% since 1980[71], while intensified dry seasons have facilitated the spread of wildfires[72,73]. These changes cause extensive social, economic and health

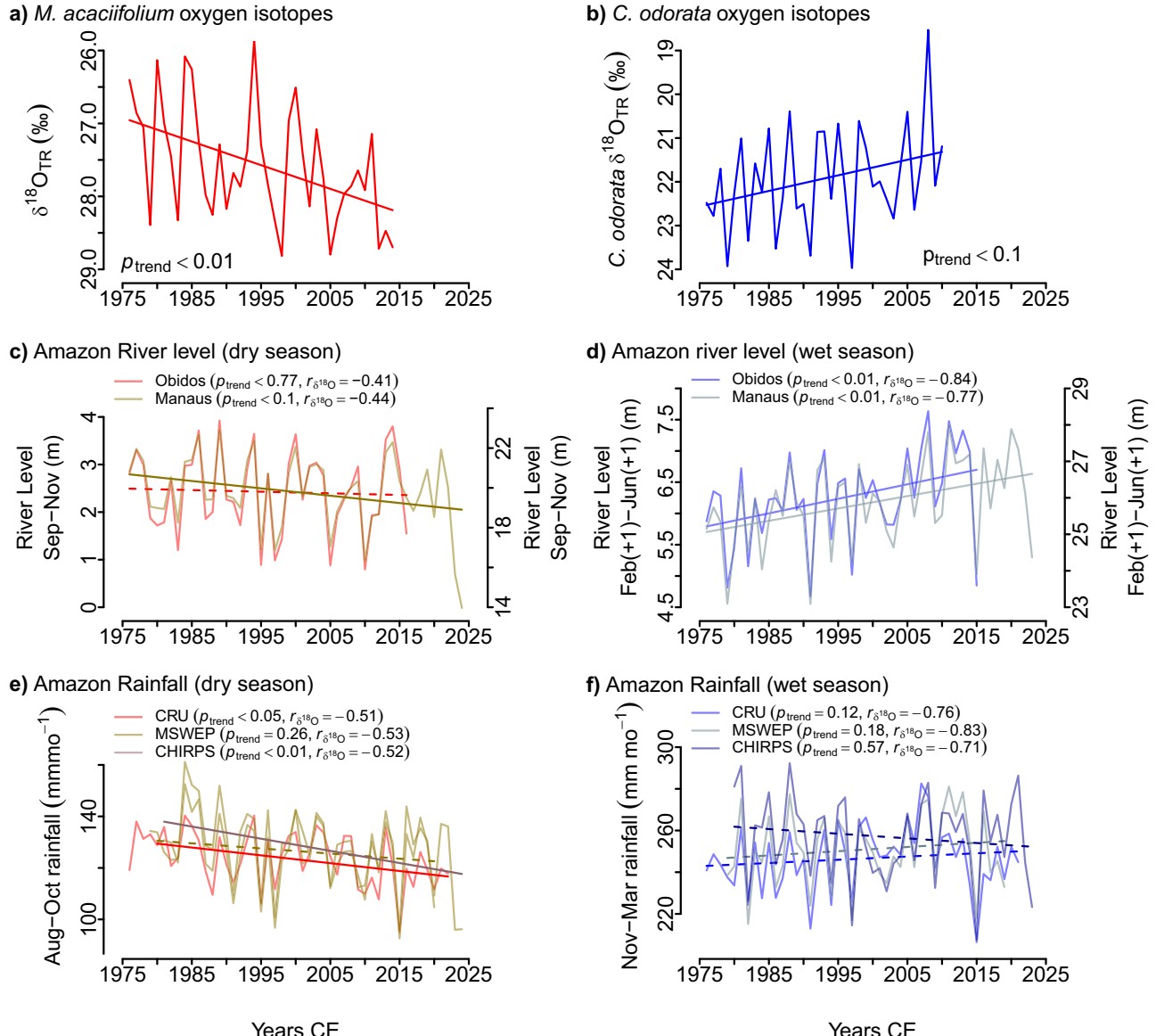

**Fig. 4 | $\delta^{18}O_{TR}$ from the flooded and terra firme sites reflect interannual variation and long-term trends in precipitation and river level during the dry and wet seasons, respectively.** Time series of (**a**, **b**) oxygen isotopes in tree rings from floodplain and terra firme trees, respectively. (**c**–**f**) hydrological variables for the peak of the dry season (left panels) and wet seasons (right panels) since 1975 C.E shwing (**c**, **d**) Amazon River levels and (**e**, **f**) Amazon-wide rainfall from CRU, MSWEP (scale on left axis) and CHIRPS (scale on right axis). Solid and dotted straight lines indicate trends that are significant within confidence intervals of 95% and 90%, respectively. The significance of trends was tested by means of linear regression, with $p$ values shown at each graph. The Pearson's correlation coefficient of the hydrological variable with oxygen isotopes from tree rings is also shown on (**d**, **e**, $r_{\delta18O}$). Note that in (**c**), the $\delta^{18}O$ axis is inverted.

impacts[74], including damages to agriculture[5], hydropower generation[75], and respiratory health[76]. For the forests, floods, droughts and fires may lead to widespread tree mortality, both in floodplains and non-flooded areas[77,78], and impact on forest biodiversity[79], structure[80], and resilience[1].

Our support for the existence of diverging long-term changes in rainfall amounts during the wet and dry seasons has important implications for proxy-based studies in this region. Commonly used proxy records such as ice-cores rarely contain seasonal climate information[26], while cave speleothems[28] usually integrate signals over periods of three or more years. Such records will conceal diverging seasonal patterns as detected here, and thus fail to provide an accurate picture of the full climate response. Tree rings do provide information about seasonal climate variability, but in the tropics such information is often restricted to the wet season[39]. The approach we used here overcomes such limitations, and exploits differences in growing seasons between species to permit seasonally explicit climate reconstructions. This approach is limited only by the maximum age of

tropical trees, which may reach up to few hundred years in these ecosystems[81].

To conclude, we find diverging time trends in these two $\delta^{18}O_{TR}$ records since ca. 1980, with increasing $\delta^{18}O_{TR}$ in the species growing during the dry season and decreasing $\delta^{18}O_{TR}$ for the species growing during the wet season. These diverging trends are indicative of approximately 5 to 7.1% per decade (15.5–22% since 1980) increase in wet season rainfall and by 1.9 to 4.4% per decade (6–13.5% since 1980) decrease in dry season rainfall at large-scale in the Amazon. Our analysis highlights that traditional climate proxies may hide seasonally diverging trends as observed here. We find no support for a general drying or general wetting of the Amazon basin and instead find an increase in the seasonal cycle. Our conclusions are largely independent from any climate data, relying only on the well-known relationship between rainfall and the $\delta^{18}O$ signals in rainfall as recorded by tree rings. The results provide a benchmark for testing the realism of climate predictions for this region and confirm a trend in seasonality that is larger than assumed so far.

These changes have severe and widespread implications for human livelihoods and ecosystems in the Amazon.

## Methods

### Experimental design

To reconstruct seasonally resolved $\delta^{18}O$ from the Amazon basin, we used tree ring records from *Cedrela odorata* from terra firme forests in the Southwestern Amazon[31,82] (10°5′S, 66°18′W), and from *Macrolobium acaciifolium* from floodplain forests in the Western Amazon[37] (74°05′ W, 4°29′ S, Fig. 1). *C. odorata* grows during the local wet season, defined as the period of the year with rainfall exceeding 100 mm month[-1] (which is October to April at this site)[31], and the $\delta^{18}O_{TR}$ chronology (i.e. the average of $\delta^{18}O$ records obtained from trees belonging to the same local population of trees) of this species thus reflects past variability of wet season rainfall in the Amazon. The inter-annual and decadal scale variation is in close agreement with similar chronologies obtained from trees of the same species growing at other locations in the western Amazon basin[82]. *Macrolobium acaciifolium* is a typical floodplain species, that grows only during the non-flooded (terrestrial) period[40], which is between July and November for this site. As this period coincides with the Amazon-wide dry season, the chronology reflects rainfall variability for the dry season. Though the two sites are distant from each other, the large-scale atmospheric processes associated with Rayleigh distillation during moisture transport result in wide spatial coherence of tree ring $\delta^{18}O$ even at distant locations[38]. This can be illustrated by the large overlap of the regions of moisture transport to each of our sampling sites (Supplementary Figs. 2 and 3).

Both chronologies consist of oxygen isotope ratios in the cellulose extracted chemically from the tree rings using the procedure described in ref. [83]. For *C. odorata*, 890 rings from 9 trees were separated prior to cellulose extraction, while for the *M. acaciifolium* trees we first extracted cellulose from cross-section laths and then separated 384 rings from 6 trees[84]. As these species grow in tropical warm climates which favour rapid drying of the soils and constant recharging with new rainfall[85,86], long soil water residence times are generally not expected[87,88], and we consider that trees use mostly recent rainfall as their main source water. Still, for *M. acaciifolium* trees only the middle section (of three segments) for each ring was used to avoid using wood produced at the start and end of the growing season, when river water (which has longer residence times) are close to trees' rooting zones. After extraction, tree ring cellulose samples were homogenized, freeze dried, packed into silver capsules and pyrolyzed with an Element Analyzer coupled to a Sercon 20/20 IRMS for analysis of their stable oxygen isotope ratios, with a precision of 0.2‰. All isotope ratios are expressed relative to the Vienna Standard Mean Ocean Water (VSMOW), in permille (‰) units. The mean $\delta^{18}O_{TR}$ chronologies for each species were produced by taking the mean $\delta^{18}O_{TR}$ of all trees for each calendar year. The average of Pearson's correlation coefficients of all pairwise combinations of trees in each chronology was 0.58 for *M. acaciifolium* and 0.63 for *C. odorata*.

### Estimation of precipitation changes using Rayleigh distillation model

We use the following Rayleigh distillation model[89]

$$\frac{\delta_{site} + 10^3}{\delta_{coast} + 10^3} = f^{\alpha-1} \equiv \left(\frac{N_{site}}{N}\right)^{\alpha-1} = \left(1 - \left(\frac{P-E}{N}\right)\right)^{\alpha-1} \quad (1)$$

to relate changes in the oxygen isotope values to changes in precipitation, moisture content in air entering the basin and evapotranspiration in Amazonia. Here, $\delta_{site}$ and $\delta_{coast}$ (‰) are relative deviations of isotope ratios from a standard ratio at the site and coast respectively, $P$ and $E$ (dimensions of mass per area) are the cumulated rainfall and evapotranspiration along an air parcel trajectory from the coast to the site, respectively, $N$ (mass per area) is the total amount of water of the air column when air enters the basin, and $N_{site}$ is the water mass left in the air column at

the end of the trajectory, $\alpha = 1.010744$ the water-vapor equilibrium fractionation factor for a condensation temperature of 10 °C[90] corresponding approximately to 700hPa altitude[91], and $f$ is the fraction of water vapour in the air column at the site relative to water vapor in air entering the basin, or 'remaining fraction'.

Our Rayleigh model assumes that (a) there have been no changes in the type of precipitation (i.e. convective vs stratiform) and intensity of precipitation, and (b) that temperature does not affect $\delta^{18}O$ strongly at these locations, and (c) that the isotopic ratio of vapor entering the basin does not change over time.

Using Eq. 1, the long-term change over time in $\delta^{18}O$ as a result of long-term changes in precipitation, evapotranspiration and changes in water content of air entering the basin is then given by

$$\delta_{site,f} - \delta_{site,i} = (\delta_{coast} + 10^3)\left\{\left(1 - \left(\frac{P_f - E_f}{N_f}\right)\right)^{(\alpha-1)} - \left(1 - \left(\frac{P_i - E_i}{N_i}\right)\right)^{(\alpha-1)}\right\} \quad (2)$$

Here the subscript 'i' stands for initial and the subscript 'f' for final in time (in our case 1980 and 2010). Equation 2 can be rewritten as a function of relative changes $\Delta P \equiv \frac{P_f - P_i}{P_i}$, $\Delta E \equiv \frac{E_f - E_i}{E_i}$, $\Delta N \equiv \frac{N_f - N_i}{N_i}$ using the identities $P_i = \frac{P_f}{(1+\Delta P/P_i)}$ for P and similar for E and N to obtain

$$\delta_{site,f} - \delta_{site,i} = (\delta_{coast} + 10^3)\left\{\left(1 - \frac{P_f}{N_f} + \frac{E_f}{N_f}\right)^{(\alpha-1)} - \left(1 - \left(\frac{\frac{P_f}{(1+\Delta P/P_i)} - \frac{E_f}{(1+\Delta E/E_i)}}{\frac{N_f}{(1+\Delta N/N_i)}}\right)\right)^{(\alpha-1)}\right\} \quad (3)$$

Equation 3 permits, in principle, to determine which combinations of relative changes in precipitation and evaporation are compatible with the observed difference of the heavy isotope ratio in precipitation between 2010 and 1980 inferred from our tree ring records. Nonetheless, to use Eq. 3 for this purpose we need (i) an estimate of present $\delta_{coast}$ and (ii) estimates of the ratios $\frac{P_f}{N_f}$, $\frac{E_f}{N_f}$ and their uncertainties. We estimate the former using the observations at Belém from Global Network of Isotopes in Precipitation (GNIP), for which rainfall $\delta^{18}O$ data is available for the years of 2014 and 2015. From these years, we used the months from August to October for the peak dry season and from November to March for the peak wet season. These periods disregard the shoulder months from the growing season of the trees, to avoid an effect of year-to-year variability in the growing season length on our results.

To estimate the latter we express them as functions of remaining water vapor fraction, $f$, in an air parcel arriving at the site at the end of the period considered (here 2010) and a recycling ratio $r_{E:P} \equiv \frac{E_f}{P_f}$, the ratio of cumulated evaporation to cumulated precipitation along the air parcel path from the coast to the site. The ratios $\frac{P_f}{N_f}$, $\frac{E_f}{N_f}$ expressed as functions of $f$ and $r_{E:P}$ are $\frac{P_f}{N_f} = \frac{1-f}{1-r_{E:P}}$, $\frac{E_f}{N_f} = \frac{f-1}{1-\frac{1}{r_{E:P}}}$. $f$ we estimate from observed $\delta_{site} - \delta_{coast}$ at the end of the period considered, inverting Eq. 1:

$$f = 1 - \frac{P_f}{N_f} + \frac{E_f}{N_f} = \left(1 + \left(\frac{\delta_{site} - \delta_{coast}}{\delta_{coast} + 10^3}\right)\right)^{1/(\alpha-1)} \quad (4)$$

For our best estimate we used $r_{E:P} = 0.325$ for the wet season and $r_{E:P} = 0.4$ for the dry season[45]. To estimate $\delta_{site} - \delta_{coast}$ we used measurements made at the GNIP stations Iquitos (Peru) for $\delta_{site}$ and Belém (Brazil) for $\delta_{coast}$, and data for Riberalta (Bolivia), from a monitoring led by the authors of this study (Brienen and Gloor). For details see Supplementary Table 1 in supplemental information.

### Rayleigh model uncertainty analysis

In order to obtain plausible ranges of relative changes of $P_f$ and $E_f$ we vary estimates of using two different estimates of $\delta_{site} - \delta_{coast}$ (one based on observations from GNIP, and one based on a isotope enabled version of the

Hadley Climate model HadAM3[92], and by varying $\alpha$ over a plausible condensation temperature range. Separately we also vary $r_{E:P}$ by ±10%.

As we know that tropical Atlantic sea surface temperature increased by 0.5 °C (80–15 W, 8–22 N, HadSST data from ref. [93]) we also examined the sensitivity of our best estimate of the attribution of the observed isotopic time trends to these changes. We did this by varying the water vapor content and the isotopic ratio of incoming air (see Supplementary Discussion for details).

## Statistical analyses

We related the $\delta^{18}O_{TR}$ chronologies to large-scale precipitation obtained from three different gridded datasets, Climate Research Unit (CRU TS 4.07)[94], from CHIRPS[95] and from MSWEP[96]. We used CRU as it is based on station data. CHIRPS and MSWEP are more recent products that combine both station and satellite data and might provide better spatial representation. Large-scale rainfall was obtained by averaging rainfall over the Amazon Basin region during the trees' respective growing seasons (Fig. 1a, gray outline[97]). To assess the regions of influence of trends in both chronologies, we calculated for both sites the daily air mass trajectories using HYSPLIT[98,99] (Supplementary Fig. 3ab) and then calculated the latitudinal range covered by >95% of all trajectories for every 0.25 degrees longitude, using a smoothing spline to remove noise. This provided the northern and southern limits of the regions of moisture transport to each sampling site (Fig. 1a). We then averaged rainfall within this region (upwind region rainfall). The rainfall series obtained by doing so is highly correlated with the rainfall time series obtained for the Amazon Basin region (Supplementary Fig 2).

We also correlated the $\delta^{18}O_{TR}$ with Amazon River level recorded at Obidos, approximately 600 km from the Amazon delta. River levels at this location corresponds to nearly 80% of the water drained within the Basin[100], thus providing a second measure of large-scale hydrology within the Basin. The presence of trends in the $\delta^{18}O$ chronologies and climate data over the past 30 years was assessed using linear regression analyses and t-tests.

## Reporting summary

Further information on research design is available in the Nature Portfolio Reporting Summary linked to this article.

## Data availability

Oxygen isotopes data that support the findings of this study are available in Figshare with the identifier https://doi.org/10.6084/m9.figshare.29070431.

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

## Acknowledgements

We gratefully acknowledge the NOAA Air Resources Laboratory (ARL) for the provision of the HYSPLIT transport and dispersion model and/or READY website (https://www.ready.noaa.gov) used in this publication. This work is supported by: Conselho Nacional de Desenvolvimento Científico e Tecnológico—Brazil grant 457423/2014-5 (JS, BBLC). Conselho Nacional de Desenvolvimento Científico e Tecnológico—Brazil grant 311874/2017-7 and 311247/2021-0 (JS). Conselho Nacional de Desenvolvimento Científico e Tecnológico—Brazil grant 207400/2014-8 (BBLC). Research Councils UK/ Natural Environment Research Council grant NE/L0211160/1 (RJWB). Research Councils UK/Natural Environment Research Council grant NE/ M02203X/1 (RJWB, MG). Research Councils UK/Natural Environment Research Council grant NE/K01353X/1 (MG). Research Councils UK/ Natural Environment Research Council grant IP−1424-0514 (MG, RJWB). Research Councils UK/Natural Environment Research Council grant IP −1314-0512 (MG, RJWB). Fundação de Amparo à Pesquisa do Estado do Amazonas – FAPEAM/Brazil grant 146/2015 (JS, BBLC). European Research Council grant 771492 (JCAB). JB was supported by a UK Research and Innovation Future Leaders Fellowship (grant ref: MR/X034097/1).

## Author contributions

Conceptualization: Bruno B L Cintra, Roel Brienen, Manuel Gloor, Jessica Baker, Jochen Schöngart, Arnoud Boom. Data Analyses: Bruno B L Cintra, Roel Brienen, Jessica Baker, Manuel Gloor, Kanhu Pattnayak. Laboratory analyses: Bruno B L Cintra, Arnoud Boom, Jessica Baker, Jochen Schöngart, Santiago Clerici. Writing the paper: Bruno B L Cintra, Roel Brienen, Manuel Gloor, Jessica Baker. Final version review and contribution: all authors.

## Competing interests

The authors declare no competing interests.
