## [Transparent Peer Review file · Communications Earth & Environment]

Tree ring isotopes reveal an intensification of the hydrological cycle in the Amazon

Corresponding Author: Dr Bruno Barcante Ladvocat Cintra

Version 0:

Decision Letter:

Dear Dr Barcante Ladvocat Cintra,

Your manuscript titled "Independent evidence of hydrological cycle intensification in the Amazon from tree ring isotopes" has now been seen by 3 reviewers, whose comments are appended below. You will see that they find your work of some potential interest. However, they have raised quite substantial concerns that must be addressed. In light of these comments, we cannot accept the manuscript for publication, but would be interested in considering a revised version that fully addresses these serious concerns.

In revision, please address the following editorial threshold:

* Provide a compelling finding that the hydrological cycle in Amazon intensify, including representative data and potential variables affecting the precipitation change, as specifically suggested by Reviewers #2 and #3.

We hope you will find the reviewers' comments useful as you decide how to proceed. If you choose to take up this option, please either highlight all changes in the manuscript text file, or provide a list of the changes to the manuscript with your responses to the reviewers.

When resubmitting, please provide a point-by-point response to the reviewers' comments. Please submit your responses as a separate file, distinct from your cover letter where you can add responses to the Editors' comments that you do not want to be made available to the reviewers. Word files are preferred. We recommend that any figures, tables or graphs that are included in the response to reviewers are also included in the main article or Supplementary Information.

If the revision process takes significantly longer than three months, we will be happy to reconsider your paper at a later date, as long as nothing similar has been accepted for publication at Communications Earth & Environment or published elsewhere in the meantime.

Please use the following link to submit your revised manuscript, point-by-point response to the reviewers' comments with a list of your changes to the manuscript text (which should be in a separate document to any cover letter), a tracked-changes version of the manuscript (as a PDF file) and any completed checklist:

Link Redacted

Please do not hesitate to contact us if you have any questions or would like to discuss the required revisions further. Thank

you for the opportunity to review your work.

Best regards,

Mengjie Wang
Associate Editor
Communications Earth & Environment
@CommsEarth

EDITORIAL POLICIES AND FORMAT

If you decide to resubmit your paper, please ensure that your manuscript complies with our editorial policies and complete and upload the checklist below as a Related Manuscript file type with the revised article:

Editorial Policy Policy requirements
(Download the link to your computer as a PDF.)

- Behavioural and social science
- Ecological, evolutionary & environmental sciences
- Life sciences

<https://www.nature.com/documents/nr-reporting-summary.zip>

For your information, you can find some guidance regarding format requirements summarized on the following checklist: (<https://www.nature.com/documents/commsj-phys-style-formatting-checklist-article.pdf>) and formatting guide (<https://www.nature.com/documents/commsj-phys-style-formatting-guide-accept.pdf>).

REVIEWER COMMENTS:

Reviewer #1 (Remarks to the Author):

Review of 'Independent evidence of hydrological cycle intensification in the Amazon from tree ring isotopes' by Bruno B. L. Cintra et al.

In this manuscript, the authors present an analysis of tree ring oxygen isotope records to show that the Amazon hydrological cycle has intensified since 1980. This study demonstrates scientific novelty that the wet season is becoming wetter while the dry season is becoming drier in the Amazon, contrasting with the general wetting or drying trend as discussed in previous studies. While this "wet season gets wetter, dry season gets drier" phenomenon has been studied in other regions, it has not been extensively explored in the Amazon. This is also an important question to study as changes in the Amazon rainforest's hydrological cycle could push the region toward a tipping point, with far-reaching implications for global climate and biodiversity.

Another novel aspect of this study is the use of different tree species to capture the signals from different seasons. By comparing $\delta^{18}\text{O}$ trends from trees growing in varying environments and employing a Rayleigh distillation model, the authors identify opposing trends—wet season rainfall increasing and dry season rainfall decreasing. These tree ring isotope records offer unique evidence that the seasonality of the Amazon hydrological cycle is strengthening, which aligns with existing climate records. The study also quantifies the wetting/drying trend and conducts sensitivity tests, exploring how variations in different parameters affect the estimates. Overall, the findings offer valuable insights into regional climate change patterns. The scientific analysis is generally robust, though some figure presentations and discussions would benefit from further improvement. The writing is mostly clear, with minor grammatical and stylistic adjustments suggested for smoother phrasing and enhanced clarity. I believe this will be an important contribution to this journal. Overall, I recommend that the paper undergo minor revisions before publication in Communications Earth & Environment.

General comments:

- The definition of wet and dry seasons seems inconsistent in the data analysis: e.g., Line 279&286: wet season: Oct to Apr; dry season: May to Oct; Line 353: dry season: June to October; wet season: November to March; Figure 4 & Supplementary Figure 2b-g: wet season: Jan-Mar; dry season: Aug-Oct). Please try to be consistent or explain why they are defined by different months.
- The tree ring $\delta^{18}\text{O}$ values range between 20-30‰, while rainfall $\delta^{18}\text{O}$ values in the region are usually around -5‰. This suggests that physiological effects can alter $\delta^{18}\text{O}$ values by as much as ~30‰ from rainfall to tree ring cellulose. Will these physiological effects vary across seasons or years, potentially impacting the observed trends and the reliability of tree ring isotopes as proxies for rainfall isotopes? Please elaborate more on this, in addition to the discussion of leaf evaporation.
- In Figure 4b&d, before the 1980s, there was actually an increase in $\delta^{18}\text{O}_{\text{TR}}$ and a decrease in water level, which contrasts with the trend after 1980. What are some potential reasons for this change?
- There may be uncertainty in ring counting for chronologies. In tropical regions, the seasonality is less distinct compared to

higher latitude regions, which raises concerns about accurately counting the annual rings. How did the authors account for the possibility of missing or overcounting rings?

Specific comments:

Main text:

- Line 96: 'should allow' -> 'allow for'
- Line 125-127: better to avoid single-sentence paragraphs, consider elaborating more details or combine with the previous paragraph.
- Line 147: it may not be accurate to use the equation for 'saturated vapour pressure' because the sea surface RH is not always 100%.
- Line 215: 'contrasting' sounds like the trends regionally are opposite to your observations. Consider rephrasing.
- Line 263-264: if no other proxy data indicated drying during the dry season of the Amazon basin, what about rainfall or river level observation data? For example, water gauge data from Manaus and Obidos should show opposite trends in seasonal water level changes, even though it may be delayed by several months.
- Line 383-384: seems like these are three datasets instead of two
- Figure 1, Line 666-668: remove 'Color shading areas indicate the upwind regions of moisture transport from each site, derived from the HYSPLIT back trajectories (see Supplementary Figure 3 and Methods section).' as the color shading is not shown in this figure.
- Figure 1c: it is confusing when 'local' in two subfigures actually refers to two different sites. Please label the site more clearly in the axis title.
- Figure 2: in the figures, 'Apr-Jun' occurs at both the beginning and the end of the x-axis, but why with different corresponding r-values?
- Figure 3b: should the green label '+f' be '+E'?
- Figure 3ab: In figure a), it looks like that $\delta^{18}O$ is lowest at 'X3' (downstream, dark blue) and highest at 'X1' (upstream, light blue), but figure b seems to show the opposite relationship.
- Figure 4ef: some lines overlap with the legend.
- Line 706: '(right period)' -> '(right panel)'
- Line 707: 'Solid straight and dotted lines' -> 'Solid and dotted straight lines'
- Lines 705-707 and Lines 711-713 are repetitive. Please combine the information for conciseness.

Supplementary materials:

- Line 48: 'evapotranspirarion' -> 'evapotranspiration'
- Line 92: 'footprins' -> 'footprints'
- Supplementary Figure 2a: It would be helpful to mark the sampling sites' locations on the map.
- Supplementary Figure 3: if the number in the legend in figure c&d indicates from which sea locations in (figure a&b, black diamonds) the $\delta^{18}O$ was calculated from, then should figure c and d be swapped?
- Supplementary Figure 5: it is not so clear which axis corresponds to which line. It would be helpful to add 'Amazon-wide rainfall from CRU, MSWEP (scale on left axis) and CHIRPS (scale on right axis).' as explained in Figure 4.
- Supplementary Table 3&4: add units for the $\delta^{18}O$ values
- Supplementary Table 4: The period of observations at the Bolivia site is different from the model simulation period. Does this affect the comparison?

Reviewer #2 (Remarks to the Author):

Dear editor and authors, I carefully reviewed the manuscript entitled "Independent evidence of hydrological cycle intensification in the Amazon from tree ring isotopes". Based on the opposing growing season and precipitation sensitivity of two species growing on distinct ecosystems, this study propose that there is a positive (negative) trend in precipitation for the wet (dry) season for the last few decades. The authors base their study in previous evidence pointing to tree-ring isotopes of *Cedrela odorata* and *Macaranga acaciifolia* as reliable indicators of wet and dry season Amazon precipitation, respectively. This is a creative approach to answer a relevant scientific question with seemingly contradictory results in past research, which makes the manuscript an important and valuable contribution to the scientific community. The manuscript is very well written and the figures are good.

However, I found some caveats, and therefore suggest major revisions:

One of my concerns is related to the spatial variability of precipitation amounts and trends throughout the Amazon. The authors are only using trees from one 'terra firme' and one 'floodplain' site to draw conclusions for the entire Amazon basin. To strengthen their results the authors could explore other $\delta^{18}O$ records from the Amazon basin (e.g. Baker et al. 2015 <http://dx.doi.org/10.1016/j.gloplacha.2015.09.008>; Ortega-Rodriguez et al. 2023 <https://doi.org/10.1016/j.scitotenv.2023.162064>; Ballentyne et al. 2011 <https://journals.ametsoc.org/view/journals/eint/15/5/2010ei277.1.xml>; etc). An additional methodology would be to spatially map the correlations with gridded precipitation data (eg. CRU, CHIRPS, GPCC) to see how spatially consistent are the correlations with tree-ring $\delta^{18}O$, and thus how consistent are the proposed precipitation trends, throughout the Amazon basin.

My second concern is related to the chosen period for calculating the trends. The authors seem to have used the *Cedrella odorata* $\delta^{18}O$ record (10°5'S, 66°18'W) published in Baker et al. (2022) that extends 200 years (<http://dx.doi.org/10.1016/j.gloplacha.2015.09.008>). However, while they calculated the dry season trend since 1970, they

only used the *Cedrella odorata* data from 1980 to calculate the wet season trend. Why don't they use the Baker et al. record before 1980 to calculate the trend? It's possible that the reported trend will change because of low $\delta^{18}\text{O}$ values in the 1970s. Similarly, $\delta^{18}\text{O}$ chronologies reported in recent articles (e.g. Ortega-Rodriguez et al. 2023 <https://doi.org/10.1016/j.scitotenv.2023.162064>) show an increasing trend in $\delta^{18}\text{O}$ and decreasing Amazon rainfall since 2010, thereby potentially disagreeing with the proposed increasing wet season trend in the Amazon basin.

To further improve these caveats I would like to see other tree-ring records mentioned in this study. In particular, what do the tree-ring width records add (or compare) to the story? For instance, the authors could check:

Granato-Souza et al. 2019 2020 (<http://dx.doi.org/10.1007/s00382-018-4227-y>)

Granato-Souza et al. 2020 (<https://doi.org/10.1029/2020GL087478>).

Humanes-Fuente et al. 2020 (DOI= 10.1029/2020JD032565)

Oelkers et al. 2023 (<https://doi.org/10.1016/j.dendro.2023.126090>)

Stahle et al. 2020 (<https://iopscience.iop.org/article/10.1088/1748-9326/ababc6>)

Vargas et al. 2022 (DOI= 10.1016/j.gloplacha.2022.103791)

Please find below some specific comments/ suggestions on Figures or text.

Fig. 1: Line 666: which are the color shading areas?

Fig. 2: Why correlating with entire Amazon basin rainfall and not the local region?
What about correlations between climate tree-ring width? Do these tell a similar story?

Fig. 3: panel d + e show different time period (1970-2014 versus 1980-2010), why?

Line 109:110: how do you define growing season? It seems that significant correlations are lagged compare to your defined growing season.

Line 208: you can compare with CHIRPS that is also based on remote sensing, thus less biased in terms of spatial availability of weather stations.

Fig S3: Vertical lines in panel d indicate period Nov-Mar, but panel f depicts Jan-Mar. Shouldn't these represent the same months?

Fig. S4: do the upwind regions correspond to the ones shown in Fig. S2?

Line 219-222: A map representing CRU and CHIRPS precipitation trends would help to better understand whether there differences / similarities in precipitation trends in space.

Line 225: Some authors have also reported overall increasing trends in the last 4 decades for the Amazon basin (e.g. <https://doi.org/10.3390/w12051244>).

Line 299: where the three segments of equal size? how were they measured?

Line 313: (e.g. 122)?

L383-385: any other observational data apart from Obidos discharge? meteorological station data?
To what extent is the local precipitation represented by $\delta^{18}\text{O}$ in both sites?

Supplementary Table 2. I didn't find all the equation parameters described.

Reviewer #3 (Remarks to the Author):

Cintra et al. investigated precipitation change in the wet and dry season over the Amazon basin in the past four decades with tree ring isotope as a tool. Tree-ring isotopes from terra firme and floodplain were used to estimate precipitation change in the wet and dry season, respectively. A simple Rayleigh distillation model indicated that wet season precipitation increased by 14-25% (terra firme) and dry season rainfall decreased by 17-25% (floodplain), based on the isotopic signals in tree ring. The authors concluded that hydrological cycle intensification due the precipitation range amplification in the two sites.

I found the studied subject really interesting. Land use and cover change and climate change substantially reshaped the spatial pattern of hydrological cycle in the Amazon. Hence, it is critical to investigate the regime change in hydrological cycle and its potential impacts. However, after reading the manuscript I have a number of concerns, questions, and remarks on the methodology and conclusion, which I think extra analyses or evidences are necessary to support their results.

Major comments:

1. Contrary to temperature, changes in precipitation have a large spatial variations, (Haghtalab et al., 2020) (also noted in L220-222), although the two sites are characterized by similar seasonal cycle. The two sites used in this study are about 1000 km apart. Although previous studies have indicated some regional dry-season drying and wet-season wetting over the Amazon basin, it is problematic to use one site as the representative of dry season and the other one as the wet season and use their precipitation range change as the indicator of hydrological cycle intensification. Additional analyses are necessary to indicate the two sites also have similar regional wet- and dry-season precipitation trend with observational precipitation dataset (e.g. the spatial pattern). Only in this way their conclusion is on the ground.

2a. A large number of studies have shown the important contribution of basin-scale moisture recycling (mainly evapotranspiration) to regional precipitation, including the isotopic method. Recent moisture-tracking model indicates the contribution is at the level of about 30% (Staal et al., 2018; van der Ent et al., 2010). Hence, regional vegetation or evapotranspiration may play a role in precipitation change. This is not consistent with this result based on tree-ring isotope (L162-163). The contribution may not be simply linear or at a constant ratio to evapotranspiration. In area with abundant air moisture such as the tropics, a percentage change in evapotranspiration may result in several times of percentage change in precipitation (Baudena et al., 2021). Hence, the contribution from moisture recycling (evapotranspiration) to precipitation may be underestimated in this study, given the large changes in land use and regional warming, as noted in the abstract and also previous studies.

2b. The changes in tree-ring isotopes are impacted by various factors. The authors have presented them in table 1. The magnitude of individual factor-induced precipitation change is comparable to each other. I am confused how to determine the impact of land-use changes and global warming on precipitation. In addition, what is the uncertainty of the modeled precipitation change?

Minor comments:

L29: let the reader to know the two sites are not in one place or nearby, e.g. add the distance or longitude, latitude information.

L30: a revised Rayleigh distillation model...?

L44: the impact of deforestation on precipitation is related to the spatial scale, e.g. Spracklen et al. (2018), Lawrence and Vandecar (2014).

L46: add relevant refs, e.g. Cui et al. (2020)

L132: Figure 3a or 3c?

L155: Water balance-based method indicates that the Amazon are experiencing substantial evapotranspiration reduction (Ma et al., 2024), so the assumption is not valid.

L156-159: another driver is the Earth greening, which raise evapotranspiration (Piao et al., 2020).

L188-189: In L161 the author shown that the tree-ring isotope and precipitation are not sensitive to evapotranspiration change. It is contrary to the statement here.

References

- Baudena, M., Tuinenburg, O. A., Ferdinand, P. A., & Staal, A. (2021). Effects of land-use change in the Amazon on precipitation are likely underestimated. *Glob Chang Biol*. doi:10.1111/gcb.15810
- Cui, J., Piao, S., Huntingford, C., Wang, X., Lian, X., Chevuturi, A., . . . Kooperman, G. J. (2020). Vegetation forcing modulates global land monsoon and water resources in a CO₂-enriched climate. *Nature Communications*, 11(1), 5184. doi:10.1038/s41467-020-18992-7
- Haghtalab, N., Moore, N., Heerspink, B. P., & Hyndman, D. W. (2020). Evaluating spatial patterns in precipitation trends across the Amazon basin driven by land cover and global scale forcings. *Theoretical and Applied Climatology*, 140(1-2), 411-427. doi:10.1007/s00704-019-03085-3
- Lawrence, D., & Vandecar, K. (2014). Effects of tropical deforestation on climate and agriculture. *Nature Climate Change*, 5(1), 27-36. doi:10.1038/nclimate2430
- Ma, N., Zhang, Y., & Szilagyi, J. (2024). Water-balance-based evapotranspiration for 56 large river basins: A benchmarking dataset for global terrestrial evapotranspiration modeling. *Journal of Hydrology*, 130607. doi:https://doi.org/10.1016/j.jhydrol.2024.130607
- Piao, S., Wang, X., Park, T., Chen, C., Lian, X., He, Y., . . . Myneni, R. B. (2020). Characteristics, drivers and feedbacks of global greening. *Nature Reviews Earth & Environment*, 1(1), 14-27. doi:10.1038/s43017-019-0001-x
- Spracklen, D. V., Baker, J. C. A., Garcia-Carreras, L., & Marsham, J. H. (2018). The Effects of Tropical Vegetation on Rainfall. *Annual Review of Environment and Resources*, 43(1), 193-218. doi:10.1146/annurev-environ-102017-030136
- Staal, A., Tuinenburg, O. A., Bosmans, J. H. C., Holmgren, M., van Nes, E. H., Scheffer, M., . . . Dekker, S. C. (2018). Forest-rainfall cascades buffer against drought across the Amazon. *Nature Climate Change*, 8(6), 539-543. doi:10.1038/s41558-

018-0177-y

van der Ent, R. J., Savenije, H. H. G., Schaefli, B., & Steele-Dunne, S. C. (2010). Origin and fate of atmospheric moisture over continents. *Water Resources Research*, 46(9), W09525. doi:10.1029/2010wr009127

Communications Earth & Environment is committed to improving transparency in authorship. As part of our efforts in this direction, we are now requesting that all authors identified as 'corresponding author' create and link their Open Researcher and Contributor Identifier (ORCID) with their account on the Manuscript Tracking System prior to acceptance. ORCID helps the scientific community achieve unambiguous attribution of all scholarly contributions. You can create and link your ORCID from the home page of the Manuscript Tracking System by clicking on 'Modify my Springer Nature account' and following the instructions in the link below. Please also inform all co-authors that they can add their ORCIDs to their accounts and that they must do so prior to acceptance.

Version 1:

Decision Letter:

Dear Dr Barcante Ladvoat Cintra,

Your manuscript titled "Independent evidence of hydrological cycle intensification in the Amazon from tree ring isotopes" has now been seen by our reviewers, whose comments appear below. In light of their advice, we are delighted to say that we are happy to publish a suitably revised version in *Communications Earth & Environment*, in principle, provided that you could address the concerns from reviewer #2 about the analysis of precipitation trends and the impact of growing season definitions.

We therefore invite you to revise your paper one last time to address the remaining concerns of our reviewers. At the same time we ask that you edit your manuscript to comply with our format requirements and to maximise the accessibility and therefore the impact of your work.

EDITORIAL REQUESTS:

*****Please take care to match our formatting and policy requirements. We will check revised manuscript and return manuscripts that do not comply. Such requests will lead to delays. *****

SUBMISSION INFORMATION:

OPEN ACCESS:

Communications Earth & Environment is a fully open access journal. Articles are made freely accessible on publication. For further information about article processing charges, open access funding, and advice and support from Nature Research,

please visit <https://www.nature.com/commsenv/open-access>

Link Redacted

Best regards,

Mengjie Wang

Associate Editor, Communications Earth & Environment

<https://www.nature.com/commsenv/>

Consulting Editor, Communications Sustainability

<https://www.nature.com/commssustain/>

Bluesky: @commsearth.nature.social; @commssustain.nature.com

REVIEWERS' COMMENTS:

Reviewer #1 (Remarks to the Author):

The authors have adequately addressed my questions and comments, and the ambiguous parts have been clarified in the revised manuscript. I have no further comments on the current version.

Reviewer #2 (Remarks to the Author):

Second review for "Independent evidence of hydrological cycle intensification in the Amazon from tree ring isotopes" submitted to Communications Earth and Environment.

I appreciate the efforts that the authors made to improve the manuscript and reply to the reviewer's concerns. Overall, the authors clarified most of their concerns, which improved the clarity, reproducibility, and quality of the manuscript. However, some of the reviewer's comments related to rainfall trends reported and seasons used, may need further attention to avoid subjective results. Please find below more detailed comments.

1) Comments regarding the rainfall trends and intensification of the Amazon's hydrological cycle.

- L134-135: If the authors don't use statistical tools to determine the year of changing trends, the statement in these lines is rather weak. We need to put present conditions in the long-term context to be able to say that climate has changed in the last few decades. The authors could explore running a shift detection in trends (or means) to strengthen their main conclusion stating that the Amazon's hydrological cycle has intensified, given divergent trends for the wet and dry seasons. They can try this with the longer Cedrela chronology. Otherwise, please acknowledge the caveats of choosing a subjective period for the conclusions, or better justify the selected period (1980-2010).

- Soil water enrichment. Another factor that the authors have not acknowledged is the potential evaporative enrichment of soil water before root absorption. Rain water can evaporate with high temperature and VPD, therefore increasing $\delta^{18}\text{O}$. Please take this into account or clarify what your assumptions are in this regard.

- Supplementary L47-49: "Over the analysed period in our study, Amazon temperature has increased by approximately 0.6oC, which would decrease cellulose $\delta^{18}\text{O}$ by 0.045‰ according to the temperature $\delta^{18}\text{O}$ -relationship found by reference 13." Are you referring to $\delta^{18}\text{O}$ or $\Delta^{18}\text{O}$?

- Supplementary L28-33 and Supplementary Table 2. I can't follow the conclusion that isotopic enrichment has a minor effect on long-term rainfall trends. How did the authors evaluate the evaporative enrichment and stomatal conductance (gs) responses to vapor pressure deficit? Maybe I'm a bit lost because I still don't find all the parameters in Table S2 described (view my comment on first review). E.g. What do $\delta^{18}\text{O}_{\text{air}}$, $\delta^{18}\text{O}_{\text{sw}}$, $\Delta^{18}\text{O}_{\text{e}}$, etc. mean? Please make sure you add full

names and describe all the parameters you mentioned here and throughout the manuscript.

2) Comments regarding the seasons used.

- L107-117: Growing seasons are still confusing.
- Fig. 1C: What do the vertical lines represent (continuous and dotted)? This figure looks like a good opportunity to graphically represent and help clarify the different seasons you described during your manuscript:
 - growing season,
 - wet and dry season,
 - non-flooded (terrestrial) period,
 - peak wet and peak dry season used for rainfall d18O data
 - growing seasons for fig 2
- Fig S2: peak wet/dry season does not seem to coincide with the climograms of Fig 1c (dry = Aug-Oct and wet = Nov-March). Also, Fig 3 uses a different period for the dry season (i.e. = June-Oct)
- In the responses-to-reviewers file you mentioned that “For the terra firme trees, the growing season is defined as the period with rainfall larger than 100mm month⁻¹. For the floodplain trees the growing season is defined based on the terrestrial phase from July to November. This is graphically explained in Figure 1c.” By looking at Fig. 1c this seems more like 150 mm. Please clarify this threshold and mention it in the manuscript. Also, see my previous comment on Fig 1c.
- Why, if floodplain trees grow from July to November, you defined a different growing season in Fig. 2?
- In the response to reviewers you added some explanation about the extended correlations beyond the growing season for *C. odorata*. To help clarify the confusion arising when using different groups of months during the manuscript, you could add this explanation.

Minor comments:

Fig S1: What does the inset plot represent?

L764-766: This is a bit difficult to follow. Mixing 1-month with three-month periods. Please be consistent.

L132-134 ; 138-139 and 144-147: check consistency between text and figures.

L273-275: highest and lowest of which period?

Supplementary section “Assessment and quantification of uncertainties to estimates of rainfall changes inferred from trends in d18OTR”. Please clarify what period do you refer to when talking about rainfall predictions along these paragraphs.

Supplementary L58-59: “only small effects on predictions in rainfall trends”. Table 1 shows up to 13% change in rainfall prediction.

Table 1 and Supplementary L74-76: check the signs in the table. Also, changes in rainfall prediction seem to be much higher than 1%.

Supplementary Figure 8: Is panel “f” at the bottom of the figure actually “e”?

L439-440: If you refer to 1980-2010 this would be a total of 30 years

Fig. 3B: Can you explain what does the equation on the top of the figure represent and link it with the model explained in the Methods section.

Reviewer #3 (Remarks to the Author):

I applauded that the authors made additional analyses and interpretation to address my concerns. I recommend the manuscript for publication after below edit is done.

L45: Local-scale deforestation increases precipitation, whereas large-scale deforestation reduces precipitation (Spracklen et

al., 2018). So it is "Large-scale deforestation could cause reductions in precipitation...", not "...cause large-scale precipitation"

Response to reviewer comments on the manuscript “**Independent evidence of hydrological cycle intensification in the Amazon from tree ring isotopes**”, submitted to the journal Communications Earth and Environment.

*All responses to reviewer comments are in italic font. Where text from the manuscript is quoted, changes from the initial version are highlighted in **bold**.*

REVIEWER COMMENTS:

Reviewer #1 (Remarks to the Author):

Review of 'Independent evidence of hydrological cycle intensification in the Amazon from tree ring isotopes' by Bruno B. L. Cintra et al.

In this manuscript, the authors present an analysis of tree ring oxygen isotope records to show that the Amazon hydrological cycle has intensified since 1980. This study demonstrates scientific novelty that the wet season is becoming wetter while the dry season is becoming drier in the Amazon, contrasting with the general wetting or drying trend as discussed in previous studies. While this "wet season gets wetter, dry season gets drier" phenomenon has been studied in other regions, it has not been extensively explored in the Amazon. This is also an important question to study as changes in the Amazon rainforest's hydrological cycle could push the region toward a tipping point, with far-reaching implications for global climate and biodiversity.

Another novel aspect of this study is the use of different tree species to capture the signals from different seasons. By comparing $\delta^{18}O$ trends from trees growing in varying environments and employing a Rayleigh distillation model, the authors identify opposing trends—wet season rainfall increasing and dry season rainfall decreasing. These tree ring isotope records offer unique evidence that the seasonality of the Amazon hydrological cycle is strengthening, which aligns with existing climate records. The study also quantifies the wetting/drying trend and conducts sensitivity tests, exploring how variations in different parameters affect the estimates. Overall, the findings offer valuable insights into regional climate change patterns.

The scientific analysis is generally robust, though some figure presentations and discussions would benefit from further improvement. The writing is mostly clear, with minor grammatical and stylistic adjustments suggested for smoother phrasing and enhanced clarity. I believe this will be an important contribution to this journal. Overall, I recommend that the paper undergo minor revisions before publication in Communications Earth & Environment.

We thank the reviewer for the time dedicated to comment on our manuscript.

General comments:

- The definition of wet and dry seasons seems inconsistent in the data analysis: e.g., Line 279&286: wet season: Oct to Apr; dry season: May to Oct; Line 353: dry season: June to October; wet season: November to March; Figure 4 & Supplementary Figure 2b-g: wet season: Jan-Mar; dry season: Aug-Oct). Please try to be consistent or explain why they are defined by different months.

We acknowledge this wasn't entirely consistent, and we have now corrected this throughout the manuscript using the following definitions: Dry season = August to October. Wet season = November to March (Figure 1c).

In line 353, for the purpose of our calculations of precipitation changes, we disregard the shoulder months of these seasons, because year-to-year variation may slightly alter them.

With these changes, the seasons used for the model calculations (Table 1) and in Figures 4, S2 and S4 are exactly the same, using the periods shown in Figure 1c.

Following this small adjustment in the definition of seasons, we recalculated all precipitation estimates resulting in very small changes in the numbers from Table 1. These small changes do not alter any of our conclusions.

The following corrections were made in the text (changes in bold):

In lines 279&286, the definitions here refer to the local wet and dry seasons. We have added the word "local" here (line 317) to make this clear.

Lines 322-324:

*"Macrolobium acaciifolium is a typical floodplain species, that grows only during the non-flooded (terrestrial) period ¹, which is between **July and November** for this site."*

Lines 390-394:

*"From these years, we used the months from **August to October** for the peak dry season and from November to March for the peak wet season. **These periods disregard the shoulder months from the growing season of the trees, to avoid an effect of year-to-year variability in growing season length on our results.**"*

- The tree ring $\delta^{18}\text{O}$ values range between 20-30‰, while rainfall $\delta^{18}\text{O}$ values in the region are usually around -5‰. This suggests that physiological effects can alter $\delta^{18}\text{O}$ values by as much as ~30‰ from rainfall to tree ring cellulose. Will these physiological effects vary across seasons or years, potentially impacting the observed trends and the reliability of tree ring isotopes as proxies for rainfall isotopes? Please elaborate more on this, in addition to the discussion of leaf evaporation.

The difference between rainwater $\delta^{18}\text{O}$ and tree ring $\delta^{18}\text{O}$ arises from a known effect of biofractionation (see original papers that first reported this effect: Epstein et al. 1977;

DeNiro and Epstein 1981; Yakir and DeNiro 1990). The existing evidence suggests that this biofractionation is only very minorly affected by climatic effects. In most tree ring studies and tree ring isotope models it is regarded to be constant at around 27 ‰. Although there is some evidence that this fractionation is temperature dependent, the overall magnitude of the effect is small, and decreases with increasing temperatures as found in the tropics¹. The estimated effect on dry season $\delta^{18}O_{TR}$ decreases is less than 5% of the overall long-term trend. We added the following text on this to the supplementary materials:

“Changes in biochemical fractionation during synthesis of carbohydrates

The $\delta^{18}O$ of tree ring cellulose in our study is about 30 ‰ higher than the $\delta^{18}O$ of rainfall. This large difference results from isotopic enrichment of leaf water and biochemical fractionation (Epsilon bio) during the synthesis of carbohydrates and cellulose. On average, the difference between $\delta^{18}O_{TR}$ and $\delta^{18}O$ from source water is approximately 27 ‰^{5,12}. While the fractionating effect of biochemical reactions may be temperature dependent, this effect has shown to be very small in warm climates¹³. Over the analysed period in our study, Amazon temperature has increased by approximately 0.6°C, which would decrease cellulose $\delta^{18}O$ by 0.045‰ according to the temperature $\delta^{18}O$ -relationship found by reference 13. In comparison, the wet season decrease in $\delta^{18}O$ is 0.9 ‰, and any potential temperature effect of biochemical fractionation on our longer-term trends is thus very small.”

- In Figure 4b&d, before the 1980s, there was actually an increase in $\delta^{18}O_{TR}$ and a decrease in water level, which contrasts with the trend after 1980. What are some potential reasons for this change?

There is a decadal event around 1970s which is thought to be associated with a well known phase change in the Pacific Decadal Oscillation and more intense ENSO variability which caused climatic shifts globally⁵⁻⁷. This event is evident in Amazon river discharge records from Obidos, and appears as a short-lived period of hydrological intensification (Gloor et al. 2013). It occurs before the start of the period we focus on in our study, from ~1980 onwards.

- There may be uncertainty in ring counting for chronologies. In tropical regions, the seasonality is less distinct compared to higher latitude regions, which raises concerns about accurately counting the annual rings. How did the authors account for the possibility of missing or overcounting rings?

We accounted for this possibility in two ways. Firstly, by using visual and statistical cross-dating of the isotope series, we were able to reach high confidence in ring dating for these two species. We found that the R_{bar} (i.e. the average correlation between all pairwise

combinations of trees) of both chronologies is (0.58 for *M. acaciifolium* and 0.63 for *C. odorata*), indicating that inaccurate datings due to non-annual or false rings are unlikely.

Secondly, the dating of subset of rings from both records was independently verified using bomb peak radiocarbon dating, confirming the absence of false rings and accuracy of ring datings. Please see references ^{8,9} for more details.

Specific comments:

Main text:

- Line 96: 'should allow' -> 'allow for'

We have made this correction.

- Line 125-127: better to avoid single-sentence paragraphs, consider elaborating more details or combine with the previous paragraph.

We have merged this with the previous paragraph.

- Line 147: it may not be accurate to use the equation for 'saturated vapour pressure' because the sea surface RH is not always 100%.

We did not assume a 100% saturation of the sea surface humidity. What we did is the following. We assumed that changes in the incoming air column water vapour content (N_i) is linearly proportional to the ratio of saturated vapor pressures (RH_{sat}) in 2010 and 1980, such that :

$$N_{i,2020} = (RH_{sat,2020}/RH_{sat,1980}) * N_{i,1980}$$

In the text say we now clarified this by adding the following text (lines 153-154):

“Assuming the change in mass of water in atmospheric inflow is linearly proportional to the ratio of saturated vapor pressures in 2010 and in 1980... “

- Line 215: 'contrasting' sounds like the trends regionally are opposite to your observations. Consider rephrasing.

We have changed the word “contrasting” with “Diverging”.

- Line 263-264: if no other proxy data indicated drying during the dry season of the Amazon basin, what about rainfall or river level observation data? For example, water gauge data from Manaus and Obidos should show opposite trends in seasonal water level changes, even though it may be delayed by several months.

We are not entirely sure what particular statement in this section the reviewer is referring to. In the sentence on lines 262-263 (“Our analysis ... here”), we talk about the fact that proxy data in general cannot reproduce seasonally resolved trends. In the following sentence, we then provide a more general concluding statement that our results do not support general drying trends (or in fact general wetting). To make this clearer, we now changed the last sentence to (lines 301-302):

“We find no support for a general drying or general wetting of the Amazon basin, but instead find an increase in the seasonal cycle.”

Regarding the rainfall or river observational records, these trends are all shown in Figure 4 and SI Figure 5. They show the same divergence as that observed in our tree ring records. The point is however that these observational records have problems themselves (see our introduction lines 63-71, copied below), which is why we used here proxy data to independently prove the seasonal changes in the hydrological cycle.

Lines 63-71:

“(...)these efforts are hampered by a lack of meteorological data. Rain gauge stations are sparsely distributed across the basin (only one station per 150,000 km²), their operation is often discontinuous, and methodologies are changing over time^{10,11}. Remote sensing and data assimilation are becoming increasingly more accurate and reliable, but they are short and there is still disagreement between these products^{11,12}. River level data may offer a more reliable, long-term record, but the varying seasonality of river levels of the many tributaries of the Amazon River may mask the timing and strength of the changes, especially for the dry season¹³”

- Line 383-384: seems like these are three datasets instead of two

Now corrected, thanks.

- Figure 1, Line 666-668: remove 'Color shading areas indicate the upwind regions of moisture transport from each site, derived from the HYSPLIT back trajectories (see Supplementary Figure 3 and Methods section).' as the color shading is not shown in this figure.

We are sorry for this oversight. We now added the mentioned color shading to this figure. It shows the footprint of the moisture transport trajectories and explains why the oxygen isotope signals at these sites record large-scale variation in precipitation.

- Figure 1c: it is confusing when 'local' in two subfigures actually refers to two different sites. Please label the site more clearly in the axis title.

We have now added the site name above the graphs.

- Figure 2: in the figures, 'Apr-Jun' occurs at both the beginning and the end of the x-axis, but why with different corresponding r-values?

Figure 2 shows correlations of the $\delta^{18}O_{TR}$ record with the yearly climate means for three-month periods. In the first April-Jun period (left), the years of the climate time series match the ring formation years from the $\delta^{18}O_{TR}$ series. In the second April-Jun period (right), the climate year is one calendar year after the ring formation, i.e. a lagged correlation. As these are different periods they will result in different correlation coefficients.

To clarify this, we have added the following statement to the figure caption:

“The x-axis spans the full hydrological year, and therefore extends across two calendar years. Months January to May on the right-hand side of the x-axis refers to the year following the year when the ring started to be formed.”

- Figure 3b: should the green label '+f)' be '+E'?

We have corrected this typo.

- Figure 3ab: In figure a), it looks like that $\delta^{18}O$ is lowest at 'X3' (downstream, dark blue) and highest at 'X1' (upstream, light blue), but figure b seems to show the opposite relationship.

We have fixed this now – thank you for pointing out.

- Figure 4ef: some lines overlap with the legend.

We have fixed this problem now.

- Line 706: '(right period)' -> '(right panel)'

We have fixed this.

- Line 707: 'Solid straight and dotted lines' -> 'Solid and dotted straight lines'

We have fixed this.

- Lines 705-707 and Lines 711-713 are repetitive. Please combine the information for conciseness.

We have changed the caption to make it more concise.

Supplementary materials:

- Line 48: 'evapotranspirarion' -> 'evapotranspiration'

We have removed this section from the supplementary text as it was a repetition from the results of the main manuscript.

- Line 92: 'footprins' -> 'footprints'

We have fixed this typo.

- Supplementary Figure 2a: It would be helpful to mark the sampling sites' locations on the map.

We have made this addition.

- Supplementary Figure 3: if the number in the legend in figure c&d indicates from which sea locations in (figure a&b, black diamonds) the $\delta^{18}\text{O}$ was calculated from, then should figure c and d be swapped?

Yes. Actually, a and b should be swapped. We have made this change.

- Supplementary Figure 5: it is not so clear which axis corresponds to which line. It would be helpful to add 'Amazon-wide rainfall from CRU, MSWEP (scale on left axis) and CHIRPS (scale on right axis).' as explained in Figure 4.

We have made this change.

- Supplementary Table 3&4: add units for the $\delta^{18}\text{O}$ values

We have made these changes.

- Supplementary Table 4: The period of observations at the Bolivia site is different from the model simulation period. Does this affect the comparison?

The record from the Bolivian site ends in 2010. The period of the data is thus 3 years shorter, which could have a small influence on the calculations. We don't think this small difference affected the comparison.

Reviewer #2 (Remarks to the Author):

Dear editor and authors, I carefully reviewed the manuscript entitled “Independent evidence of hydrological cycle intensification in the Amazon from tree ring isotopes”. Based on the opposing growing season and precipitation sensitivity of two species growing on distinct ecosystems, this study propose that there is a positive (negative) trend in precipitation for the wet (dry) season for the last few decades. The authors base their study in previous evidence pointing to tree-ring isotopes of *Cedrela odorata* and *Macrolobium acaciifolium* as reliable indicators of wet and dry season Amazon precipitation, respectively. This is a creative approach to answer a relevant scientific question with seemingly contradictory results in past research, which makes the manuscript an important and valuable contribution to the scientific community. The manuscript is very well written and the figures are good.

We thank the reviewer for the time dedicated to comment on our manuscript.

However, I found some caveats, and therefore suggest major revisions:

One of my concerns is related to the spatial variability of precipitation amounts and trends throughout the Amazon. The authors are only using trees from one ‘terra firme’ and one ‘floodplain’ site to draw conclusions for the entire Amazon basin. To strengthen their results the authors could explore other d18O records from the Amazon basin (e.g. Baker et al. 2015 <http://dx.doi.org/10.1016/j.gloplacha.2015.09.008>; Ortega-Rodriguez et al. 2023 <https://doi.org/10.1016/j.scitotenv.2023.162064>; Ballentyne et al. 2011 <https://journals.ametsoc.org/view/journals/eint/15/5/2010ei277.1.xml>; etc). An additional methodology would be to spatially map the correlations with gridded precipitation data (eg. CRU, CHIRPS, GPCC) to see how spatially consistent are the correlations with tree-ring d18O, and thus how consistent are the proposed precipitation trends, throughout the Amazon basin.

The Cedrela record from Baker et al. (2015), complemented with more recent Cedrela data, is the one we use here. The records from the other species in Baker et al. (2015) are each based on only one tree per species which is not sufficient to produce a reliable chronology, and thus not included in our manuscript.

We hesitated to include the Ballentyne et al. (2011) records in the discussion of our paper for the following reasons. The record is based on one single Cedrela tree with two radii measured. The statistical measures indicate that cross-dating between these two records from the same single tree are comparably poor. This is in contrast to our Cedrela chronology was well replicated (with a total of 32 trees) and showed good crossdating statistics.

While the Ballentyne record is much shorter than our record, the trends over the limited overlapping period are similar to ours. Overall we remain reluctant to include the Ballentyne record in our discussion.

We have added the following paragraph which compares the Ortega-Rodriguez record with our record (lines 252-255):

*“Existing tree ring records from the Amazon provide some support for the increase in wet season precipitation suggested by our records. An extended $\delta^{18}O_{TR}$ record (1980 to 2020) from *Cedrela odorata* located in the nearby Brazilian southwest Amazon (Ortega-Rodriguez et al. 2023) shows a similar upward trend in precipitation up to 2020, despite the strong 2015 El Niño (ie dry anomaly)”*

To examine the reviewer’s concern that using data from one site to infer large scale change in hydrology in the Amazon may not be sufficient, we have made correlation maps of our records with precipitation (SI Fig 6, see below). These maps confirm that both our records are representative for precipitation – at the least for the entire upwind region of the sites (a large fraction of the Basin. We attribute this to the coherent atmospheric circulation in the Basin (trade winds entering the basin, circulation turning southwards along the Andes (low level jet)).

Supplementary Figure 6. Spatial correlation maps of the $\delta^{18}O_{TR}$ records with gridded precipitation datasets confirm the large-scale climatic footprint of the $\delta^{18}O_{TR}$ records. Color scale indicates the Pearson’s correlation coefficient. Contours show regions of significant correlations ($p < 0.1$). All correlations were computed for 1980 onwards.

My second concern is related to the chosen period for calculating the trends. The authors

seem to have used the *Cedrella odorata* d18O record (10°5'S, 66°18'W) published in Baker et al. (2022) that extends 200 years (<http://dx.doi.org/10.1016/j.gloplacha.2015.09.008>). However, while they calculated the dry season trend since 1970, they only used the *Cedrella odorata* data from 1980 to calculate the wet season trend. Why don't they use the Baker et al. record before 1980 to calculate the trend? It's possible that the reported trend will change because of low d18O values in the 1970s. Similarly, d18O chronologies reported in recent articles (e.g. Ortega-Rodriguez et al. 2023 <https://doi.org/10.1016/j.scitotenv.2023.162064>) show an increasing trend in d18O and decreasing Amazon rainfall since 2010, thereby potentially disagreeing with the proposed increasing wet season trend in the Amazon basin.

To further improve these caveats I would like to see other tree-ring records mentioned in this study. In particular, what do the tree-ring width records add (or compare) to the story? For instance, the authors could check:

Granato-Souza et al. 2019 2020 (<http://dx.doi.org/10.1007/s00382-018-4227-y>)

Granato-Souza et al. 2020 (<https://doi.org/10.1029/2020GL087478>).

Humanes-Fuente et al. 2020 (DOI= 10.1029/2020JD032565)

Oelkers et al. 2023 (<https://doi.org/10.1016/j.dendro.2023.126090>)

Stahle et al. 2020 (<https://iopscience.iop.org/article/10.1088/1748-9326/ababc6>)

Vargas et al. 2022 (DOI= 10.1016/j.gloplacha.2022.103791)

The Cedrela record from Baker et al. (2015), complemented with more recent Cedrela data, is the one we use here, spanning approximately 150 years. In response to this comment and that of other reviewers, we now show the longer series since 1900 in Figure 3. Nonetheless the focus – and the result we want to highlight for this paper is what is observed during the time period since 1980, given the extreme hydrological conditions that have been happening in the Amazon Basin in the past 40 years.

To acknowledge previous tree ring work in the Amazon and more recent oxygen isotope studies we have now included a discussion that mentions some of the referenced records (excluding those from the Andes and the one from Stahle et al. 2022 which is a pan-American study). The following paragraph was included in lines 252-269:

*“Existing tree ring records from the Amazon provide some support for the increase in wet season precipitation suggested by our records. An extended $d^{18}O_{TR}$ record (1980 to 2020) from *Cedrela odorata* located in the nearby Brazilian southwest Amazon (Ortega Rodriguez et al., 2023) shows a similar upward trend in precipitation up to 2020, despite the strong 2015 El Niño (ie dry anomaly). Tree ring-widths from south-west Amazon-Andes also capture this wetting trend locally (Oelker et al. 2023), and one study from eastern Amazon points out that climate signals in the tree rings became weaker after 1990s (Granato-Souza et al. 2019), which can be an indication that growth limitation by rainfall is decreasing, even in the absence of*

trends. Other ring-width studies from different parts of the Amazon don't seem to capture this change (Granato-Souza et al. 2020, Lopez et al. 2017, Batista and Schöngart, 2018, Schongart et al. 2005). The reasons why tree ring-width do not capture these changes may vary. Firstly, trees growing in humid conditions during the wet season may not be limited by precipitation amount, and thus don't benefit from the increase in rainfall. Secondly, tree ring width studies often focus on high-frequency variability, and may not reproduce multidecadal trends. In addition, detecting trends in ring-width series requires consideration of several ontogeny and biases due to demography or field sampling strategies, which are often not considered (Brienen et al. 2012, 2016, Bowman et al, 2013). In comparison to tree ring widths, $\delta^{18}\text{O}_{\text{TR}}$ records are superior proxies for hydroclimate as the signal in these two sites arises predominantly from variation in precipitation $\delta^{18}\text{O}$ and is thus relatively independent of plant physiological effects (Brienen et al. 2012, Cintra et al. 2019) and do not require detrending to correct for ontogenetic effects.”

Please find below some specific comments/ suggestions on Figures or text.

Fig. 1: Line 666: which are the color shading areas?

We are sorry for this oversight. We now added the mentioned color shading to this figure. It shows the footprint of the moisture transport trajectories and explains why the oxygen isotope signals at these sites record large-scale variation in precipitation.

Fig. 2: Why correlating with entire Amazon basin rainfall and not the local region?

What about correlations between climate tree-ring width? Do these tell a similar story?

We have explored this in previous publications. We found that correlations with large scale climate are much stronger than with local climate (Cintra et al. 2019, 2021, Brienen et al. 2012), and this is explained in detail in the original manuscript (now in lines 82-88, 193-203).

Specifically $\delta^{18}\text{O}_{\text{TR}}$ in tree ring cellulose records large-scale climate variability because it records primarily source water $\delta^{18}\text{O}$. Source water $\delta^{18}\text{O}$ variation itself is the result of a Rayleigh distillation rainout process along air parcel paths, determined by cumulative rainout of the heavy water along the path (ie., a Rayleigh distillation signal as shown in our Figure 3; see also (Salati et al. 1970, Baker et al. 2016, Vimeux et al. 2005, Vuille and Verner 2005). This signal is well preserved in the $\delta^{18}\text{O}_{\text{TR}}$ provided that the growing conditions are wet enough.

In contrast tree ring-widths reflect tree growth responses to local climate and light variability. For terra firme trees (e.g. *C. odorata*) this includes local rainfall (Granato-Souza et al. 2019, Brienen et al. 2022). For floodplain trees (*M. acaciifolium*) it is variation in river levels, which control the length of the terrestrial period when those trees are not flooded and are able to grow (Schongart et al. 2005). The strength of the (hydrological) climate signal from tree ring

widths is much weaker compared to the $\delta^{18}\text{O}$ records. An analysis of the local climate signals from ring-widths is outside the scope of our study.

Fig. 3: panel d + e show different time period (1970-2014 versus 1980-2010), why?
In the revised version of the manuscript we report trends for only one period, starting in 1980. This has shortened the period used to for dry season calculations, resulting in some changes to the total estimated in dry season precipitation changes. We have corrected these estimates in Table 1 and throughout the manuscript.

Line 109:110: how do you define growing season? It seems that significant correlations are lagged compare to your defined growing season.

For the terra firme trees, the growing season is defined as the period with rainfall larger than 100mm month⁻¹. For the floodplain trees the growing season is defined based on the terrestrial phase from July to November. This is graphically explained in Figure 1c.

*The observed correlations between tree ring $\delta^{18}\text{O}$ and rainfall largely match the tree's respective growing seasons. The only exception is for the correlations for the *C. odorata* record extending into May and June. This could be due to interannual variation in the growing season length of the trees, which may explain why those correlations with periods that include the month of June are generally weaker than the with preceding periods that don't include this month (this is especially visible in the correlations with the CRU record, Figure 1f).*

To make this more accurate with what is shown in figure 2e-g, we have changed this sentence indicating that the peak correlations go until May for this species (lines 110-113, changes in bold):

*"Peak rainfall correlations for the floodplain $\delta^{18}\text{O}_{\text{TR}}$ chronology occurred from August to November, the main growing season for *M. acaciifolium* (Figure 2a-c), while the terra firme $\delta^{18}\text{O}_{\text{TR}}$ chronology shows peak correlations from January to **May**, coinciding with the main growth season for *C. odorata* (Figure 2d-e)."*

The other cases when correlations lag the growing season are with the river level records at Obidos at the end of the catchment. This is expected as we explain in lines 124-127:

"Because of lag times between Amazon precipitation and river level ¹⁴, these correlations are ca. 3-5 month later than the rainfall- $\delta^{18}\text{O}_{\text{TR}}$ correlations."

The same lag is expected between Amazon precipitation and river level at Óbidos.

Line 208: you can compare with CHIRPS that is also based on remote sensing, thus less biased in terms of spatial availability of weather stations.

We have now done these calculations with all three climate products used in this study, CRU, CHIRPS and MSWEP. For CRU we have also updated the data to the version 4.07, which results in different estimates. These changes in the estimates between different versions of the same dataset and between different datasets further highlight the uncertainties in estimates of long-term climate trends from available climate data from these regions. Following these changes we have changed the text (lines 218-224, changes in bold):

“Our estimates from tree ring oxygen isotopes suggest rainfall reductions in the dry season (-1.9 to -4.4% per decade) are largely within the range of those derived from the different gridded precipitation datasets used here (CRU: -0.3%, CHIRPS: -6.3%, MSWEP: -1.6%, per decade), but imply stronger rainfall increases during the wet season (+5 to +7.1% per decade) than estimated from all three datasets (CRU : +1.6%, CHIRPS: +1.6% and MSWEP: +0.3% per decade). This discrepancy in the estimates of rainfall changes may possibly be caused by differences between the footprint of the tree ring isotope series...”

Fig S3: Vertical lines in panel d indicate period Nov-Mar, but panel f depicts Jan-Mar. Shouldn't these represent the same months?

Yes, we have made this correction. The inflow change of 8% was correctly calculated over the months from November to March.

Note that we have changed the dry season period to August to October, for coherency, following comments from Reviewer 1. This change was made to all our calculations and did not make a substantial difference to our estimates.

Fig. S4: do the upwind regions correspond to the ones shown in Fig. S2?

Yes. Upwind regions in Figure S2 were used to cut the climate data that was used for the Figure S4 in all panels that indicate “upwind regions”.

Line 219-222: A map representing CRU and CHIRPS precipitation trends would help to better understand whether there differences / similarities in precipitation trends in space.

While we do agree that this would be interesting, this type of analysis would not be simple and would inevitably require too much discussion that is outside the scope of our manuscript. This is why we prefer to refer in this paragraph to numerous publications that were focused on doing this kind of analysis.

Line 225: Some authors have also reported overall increasing trends in the last 4 decades for the Amazon basin (e.g. <https://doi.org/10.3390/w12051244>).

Studies which focus on trends in annual precipitation will most likely find trends that reflect changes from the wet season, because most of the annual precipitation falls in the wet season. We have included this reference in the discussion with the following addition to this sentence in lines (244-245):

“We find no support for previous GCM predictions for a general drying of the Amazon basin⁴⁹⁻⁵¹, or for a general wetting (da Motta Paca et al. 2020).”

Line 299: where the three segments of equal size? how were they measured?

Yes, the three segments were of equal sizes. They were measured and cut with the aid of a stereomicroscope. The cutting method resulted in a very coherent subannual signal that is consistent among trees (see Response Figure 1 below).

Response Figure 1. Tree ring $\delta^{18}\text{O}$ with sub-annual resolution obtained by separating tree rings into three segments of equal size show a highly coherent sub-annual signal among different trees, with peak $\delta^{18}\text{O}_{\text{TR}}$ values in the middle segments of each ring. Figure from Cintra et al. (2019).

Line 313: (e.g. 122)?

The number 122 is a reference. We have removed the “e.g.” and fixed the formatting and updated the numbering.

L383-385: any other observational data apart from Obidos discharge? meteorological station data?

CRU is entirely based on meteorological station data, while CHIRPS is partially based on satellite data. Apart from this, Obidos discharge is the only ground-based hydrological observation. Our $\delta^{18}\text{O}_{\text{TR}}$ are also observational.

To what extent is the local precipitation represented by d18O in both sites?

*Our analyses carried out in previous publications indicate that local precipitation has a minor / negligible influence for both sites used in this study. For both sites the correlations with local rainfall are weak compared to basin wide precipitation (r -local= -0.04 (n.s.), r -basinwide= -0.57*** for Floodplain from Peru, see Cintra et al. (2021), and r -local =-0.32 (n.s.), r -basin wide = -0.80*** for Terra Firme from Bolivia, see Brienen et al. (2012))*

Local climate influences $\delta^{18}O_{TR}$ only through the effects of temperature or humidity on leaf water enrichment. Potential contributions of local climate trends on the trends in $\delta^{18}O_{TR}$ were evaluated using known tree ring stable isotopes models. Results of this sensitivity analysis are described in the supplementary materials under the heading "Contribution of leaf evaporation to tree ring $\delta^{18}O$ signals".

Supplementary Table 2. I didn't find all the equation parameters described.

The parameters that were missing were e_a and e_i . They weren't listed because these are not constant. For this analysis, we obtained e_a from vapor pressure observations from CRU TS 4.04 and calculated e_i from local temperature. We have added this information in the table now.

Reviewer #3 (Remarks to the Author):

Cintra et al. investigated precipitation change in the wet and dry season over the Amazon basin in the past four decades with tree ring isotope as a tool. Tree-ring isotopes from terra firme and floodplain were used to estimate precipitation change in the wet and dry season, respectively. A simple Rayleigh distillation model indicated that wet season precipitation increased by 14-25% (terra firme) and dry season rainfall decreased by 17-25% (floodplain), based on the isotopic signals in tree ring. The authors concluded that hydrological cycle intensification due the precipitation range amplification in the two sites.

I found the studied subject really interesting. Land use and cover change and climate change substantially reshaped the spatial pattern of hydrological cycle in the Amazon. Hence, it is critical to investigate the regime change in hydrological cycle and its potential impacts. However, after reading the manuscript I have a number of concerns, questions, and remarks on the methodology and conclusion, which I think extra analyses or evidences are necessary to support their results.

We thank the reviewer for the time dedicated to comment on our manuscript.

Major comments:

1. Contrary to temperature, changes in precipitation have a large spatial variations, (Haghtalab et al., 2020) (also noted in L220-222), although the two sites are characterized by similar seasonal cycle. The two sites used in this study are about 1000 km apart. Although previous studies have indicated some regional dry-season drying and wet-season wetting over the Amazon basin, it is problematic to use one site as the representative of dry season and the other one as the wet season and use their precipitation range change as the indicator of hydrological cycle intensification. Additional analyses are necessary to indicate the two sites also have similar regional wet- and dry-season precipitation trend with observational precipitation dataset (e.g. the spatial pattern). Only in this way their conclusion is on the ground.

*We understand the concern of the reviewer but due to the physical mechanism that determines variation in rainfall $\delta^{18}\text{O}$ (recorded in tree ring $\delta^{18}\text{O}$) at western Amazon, the large distance between the sites is not a problem. The climate signals in these records are the result of large-scale rainout of heavy water during water vapour transport to the two sites which has been shown by several studies (Salati et al. 1970, Baker et al. 2016, Vimeux et al. 2005, Vuille and Verner 2005), including one recent study with a comprehensive multiproxy approach (Orrison et al. 2024, now cited in explanations from line 85). Due to this physical mechanism, variation in local climate conditions have little influence on these records as shown by the poor correlations with local climate (see response to the previous comment from Reviewer #2 and Brienen et al. 2012, Cintra et al 2021). What drives variation in these records is the climate over the footprints of the water transport to the sites. We show that these footprint overlap significantly (Fig. 1a), and are representative of Amazon-wide climate variability (Supplementary Figures 2 and 3). These large footprints are further evidenced by the new Supplementary Figure 6, where we use spatial correlation analyses to show that the respective sites record precipitation over large parts of the amazon basin for the dry season (*M. acaciifolium*) and wet season (*C. odorata*.) (see new Supplementary Figure 6 below). Such overlapping large-scale footprints confirm that the climate signal in each of these records reflect the same regional wet and dry season precipitation trends.*

Supplementary Figure 6. Spatial correlation maps of the $\delta^{18}\text{O}_{\text{TR}}$ records with gridded precipitation datasets confirm the large-scale climatic footprint of the $\delta^{18}\text{O}_{\text{TR}}$ records. Color scale indicates the Pearson's correlation coefficient. Contours show regions of significant correlations ($p < 0.1$). All correlations were computed for 1980 onwards.

2a. A large number of studies have shown the important contribution of basin-scale moisture recycling (mainly evapotranspiration) to regional precipitation, including the isotopic method. Recent moisture-tracking model indicates the contribution is at the level of about 30% (Staal et al., 2018; van der Ent et al., 2010). Hence, regional vegetation or evapotranspiration may play a role in precipitation change. This is not consistent with this result based on tree-ring isotope (L162-163). The contribution may not be simply linear or at a constant ratio to evapotranspiration. In area with abundant air moisture such as the tropics, a percentage change in evapotranspiration may result in several times of percentage change in precipitation (Baudena et al., 2021). Hence, the contribution from moisture recycling (evapotranspiration) to precipitation may be underestimated in this study, given the large changes in land use and regional warming, as noted in the abstract and also previous studies.

In our study we have considered that evapotranspiration recycles around 40% of total dry season rainfall and 32.5% of total wet season rainfall (Baker et al. 2022), which is a higher percentage than considered in Staal et al. (2018). These numbers are stated in line 404, as $R_{E::P}=0.4$ and $R_{E::P}=0.325$ for dry and wet seasons, respectively.

In lines 162-163, we were not referring to the effects of evapotranspiration on rainfall. We are referring to the effect of changes in evapotranspiration on the Rayleigh distillation, and how this could affect our estimates of rainfall changes.

To highlight and clarify this, we have added the following sentences (in bold) to that paragraph (lines 162-180):

*“We further assumed there are no changes in the total evapotranspiration of the forest. **In our estimates, we considered that evapotranspiration recycles 40% of total dry season rainfall and 32.5% of total wet season rainfall**¹⁵. Deforestation or CO₂-induced stomatal closure may result in decreases in evapotranspiration¹⁶, while warming induced vapour pressure deficit may increase evaporative demand¹⁷. These opposing effects may offset each other¹⁸, which could explain why no changes in evapotranspiration are evident within a variety of dataset types and sources for the Amazon¹⁹. While the extent of shifts in evapotranspiration at large-scale in the Amazon remains uncertain¹⁷, **recent evidence suggests changes that Amazon evapotranspiration may have decreased by approximately 6% in the past 30 years**²⁰. **Changes in the contribution of total evapotranspiration to rainfall can influence the degree of Rayleigh distillation effect during moisture transport (Figure 3b) and thereby add uncertainty to our estimates (Eqs 3 and 4). Therefore, we conduct a sensitivity analysis to assess variations in our estimates under different scenarios of changes in evapotranspiration during the analysed period. This sensitivity analysis shows that our estimates of precipitation changes are not very sensitive to the effects of changes in evapotranspiration on the Rayleigh distillation: the Rayleigh model shows that a 1-1.5% change in evapotranspiration per decade has only a minor (± 0.5 to 0.8% per decade) effect on our estimates of precipitation changes for the two seasons (Figure 3, Table 1, case 5). Our analysis further shows that these estimates are insensitive to assumed condensation temperature of water vapour in the clouds (Table 1, case 6).”***

2b. The changes in tree-ring isotopes are impacted by various factors. The authors have presented them in table 1. The magnitude of individual factor-induced precipitation change is comparable to each other. I am confused how to determine the impact of land-use changes and global warming on precipitation. In addition, what is the uncertainty of the modeled precipitation change?

In this analysis presented in Table 1, we assess how sensitive our $\delta^{18}O_{TR}$ derived estimates for dry and wet season precipitation changes are to the model assumptions and to observed climatic changes. We like to emphasize that this analysis does not specifically identify what is driving the observed changes in large-scale precipitation in the Amazon wet and dry seasons.

We directly assessed temperature effects by changing the water-vapour equilibrium fractionation (α) in our Rayleigh model according to a 1oC Amazon temperature increase between 1980 and 2010 (Case 4, Table 1). Effects of land-use changes could not be assessed

directly, but we did assess how altering total evapotranspiration vegetation between 1980 and 2010 (Case #5) would affect our estimates of precipitation change.

The total uncertainty of modelled precipitation change is difficult to assess, but could be presented by the range of reported changes (ie dry season -5.8% to -13.5%, wet season: 15.5 to 20.7% change in rainfall) and could be seen as the lowest to highest values reported.

Minor comments:

L29: let the reader to know the two sites are not in one place or nearby, e.g. add the distance or longitude, latitude information.

We have added this information as follows (line 29, changes in bold)

*“Opposing $\delta^{18}O_{TR}$ trends in the records from terra firme and floodplain trees **from distinct sites (approximately 1000km apart) in the Western Amazon** indicate rainfall amounts increased during the wet season and decreased during the dry season.”*

L30: a revised Rayleigh distillation model...?

We wouldn't call it a revised Rayleigh distillation model. We used the known Rayleigh distillation model for two points in time, and this required some parameterization. But all the logic is still from the original model.

L44: the impact of deforestation on precipitation is related to the spatial scale, e.g. Spracklen et al. (2018), Lawrence and Vandecar (2014).

We agree, and have corrected the line to specify that we refer to deforestation causing reductions in large-scale precipitation (lines 44-45, changes in bold):

*“Deforestation could cause reductions in **large-scale** precipitation...”*

L46: add relevant refs, e.g. Cui et al. (2020)

We have added this reference to the text.

L132: Figure 3a or 3c?

Figure 3c. We have made this correction.

L155: Water balance-based method indicates that the Amazon are experiencing substantial evapotranspiration reduction (Ma et al., 2024), so the assumption is not valid.

We agree and acknowledge that this assumption is a limitation, and we therefore conduct

the uncertainty analysis presented in Table 1 to assess if our estimates are sensitive to changes in evapotranspiration. This sensitivity analysis considers an evapotranspiration change of 5% similar to those implied in the referred study from Ma et al. 2024. They estimate annual evapotranspiration for the Amazon basin in the order of 1200mm yr⁻¹ (with large uncertainties, see their Figure 4a), and estimate that this has decreased by 2.5mm yr⁻²(their Figure 6), which corresponds to approximately 6% reduction over 30 years. Our sensitivity analysis indicates that an evapotranspiration change of this magnitude would have a very small effect on our estimates (see lines 168–180). Nonetheless, we do consider this effect when reporting the range of uncertainty in our calculations. We have now cited the referred study in this paragraph.

L156-159: another driver is the Earth greening, which raise evapotranspiration (Piao et al., 2020).

We have now cited this study in this paragraph.

L188-189: In L161 the author shown that the tree-ring isotope and precipitation are not sensitive to evapotranspiration change. It is contrary to the statement here.

In the previous line 161, our conclusion was that our precipitation trend estimates based on tree ring $\delta^{18}O_{TR}$ are relatively insensitive to changes in evapotranspiration. But that does not mean that trends in the observed precipitation changes itself cannot be not influenced by or be resulting from changes in evapotranspiration. These are two separate things, and we like to emphasize that we cannot make any inferences about changes in evapotranspiration in this study. For clarity we have now removed the word “evapotranspiration” from this sentence as it was not needed in this place. It now reads as follows (lines 203-205, changes in bold):

*“The **diverging** long-term trends in $\delta^{18}O_{TR}$ **are therefore interpreted to** reflect opposing seasonal trends in precipitation for the same central region of the Amazon basin.”*

References

- Baudena, M., Tuinenburg, O. A., Ferdinand, P. A., & Staal, A. (2021). Effects of land-use change in the Amazon on precipitation are likely underestimated. *Glob Chang Biol*. doi:10.1111/gcb.15810
- Cui, J., Piao, S., Huntingford, C., Wang, X., Lian, X., Chevuturi, A., . . . Kooperman, G. J. (2020). Vegetation forcing modulates global land monsoon and water resources in a CO₂-enriched climate. *Nature Communications*, 11(1), 5184. doi:10.1038/s41467-020-18992-7
- Haghtalab, N., Moore, N., Heerspink, B. P., & Hyndman, D. W. (2020). Evaluating spatial

patterns in precipitation trends across the Amazon basin driven by land cover and global scale forcings. *Theoretical and Applied Climatology*, 140(1-2), 411-427. doi:10.1007/s00704-019-03085-3

Lawrence, D., & Vandecar, K. (2014). Effects of tropical deforestation on climate and agriculture. *Nature Climate Change*, 5(1), 27-36. doi:10.1038/nclimate2430

Ma, N., Zhang, Y., & Szilagyi, J. (2024). Water-balance-based evapotranspiration for 56 large river basins: A benchmarking dataset for global terrestrial evapotranspiration modeling. *Journal of Hydrology*, 130607. doi:<https://doi.org/10.1016/j.jhydrol.2024.130607>

Piao, S., Wang, X., Park, T., Chen, C., Lian, X., He, Y., . . . Myneni, R. B. (2020). Characteristics, drivers and feedbacks of global greening. *Nature Reviews Earth & Environment*, 1(1), 14-27. doi:10.1038/s43017-019-0001-x

Spracklen, D. V., Baker, J. C. A., Garcia-Carreras, L., & Marsham, J. H. (2018). The Effects of Tropical Vegetation on Rainfall. *Annual Review of Environment and Resources*, 43(1), 193-218. doi:10.1146/annurev-environ-102017-030136

Staal, A., Tuinenburg, O. A., Bosmans, J. H. C., Holmgren, M., van Nes, E. H., Scheffer, M., . . . Dekker, S. C. (2018). Forest-rainfall cascades buffer against drought across the Amazon. *Nature Climate Change*, 8(6), 539-543. doi:10.1038/s41558-018-0177-y

van der Ent, R. J., Savenije, H. H. G., Schaefli, B., & Steele-Dunne, S. C. (2010). Origin and fate of atmospheric moisture over continents. *Water Resources Research*, 46(9), W09525. doi:10.1029/2010wr009127

Point-by-point response to reviewer comments in the study “**Independent evidence of hydrological cycle intensification in the Amazon from tree ring isotopes**”.

Reviewer comments were pasted in normal font.

*Author responses are in italic font. Where the manuscript text is quoted, **changes to the text are in bold font**.*

We thank all reviewers once again for the time dedicated to assessing our manuscript.

REVIEWERS' COMMENTS:

Reviewer #1 (Remarks to the Author):

The authors have adequately addressed my questions and comments, and the ambiguous parts have been clarified in the revised manuscript. I have no further comments on the current version.

Reviewer #2 (Remarks to the Author):

Second review for “Independent evidence of hydrological cycle intensification in the Amazon from tree ring isotopes” submitted to Communications Earth and Environment.

I appreciate the efforts that the authors made to improve the manuscript and reply to the reviewer's concerns. Overall, the authors clarified most of their concerns, which improved the clarity, reproducibility, and quality of the manuscript. However, some of the reviewer's comments related to rainfall trends reported and seasons used, may need further attention to avoid subjective results. Please find below more detailed comments.

1) Comments regarding the rainfall trends and intensification of the Amazon's hydrological cycle.

- L134-135: If the authors don't use statistical tools to determine the year of changing trends, the statement in these lines is rather weak. We need to put present conditions in the long-term context to be able to say that climate has changed in the last few decades. The authors could explore running a shift detection in trends (or means) to strengthen their main conclusion stating that the Amazon's hydrological cycle has intensified, given divergent trends for the wet and dry seasons. They can try this with the longer Cedrela chronology. Otherwise, please acknowledge the caveats of choosing a subjective period for the conclusions, or better justify the selected period (1980-2010).

We have included an analysis in the supplementary materials to show that the difference in the trends between wet and dry season as indicated by the $\delta^{18}O$ records is unprecedented (see new Supplementary Figure 7 below), and their divergence is indicative of an intensification of the hydrological cycle.

In addition to including this figure, we have added the following statement to the results section (lines 158-159):

”The divergence between the two records since mid 1970s is greater than at any other time in the period covered (Supplementary Figure 7).”

(new) Supplementary Figure 7. The divergence between the two records since mid 1970s is greater than at any other time in the period covered. $\delta^{18}\text{O}$ Trend slopes for sliding windows of 35ys in the *C. odorata* (blue, wet season) and *M. acaciifolium* (red, dry season) $\delta^{18}\text{O}$ records. Solid line indicates the difference between the trend slopes of the two records.

- Soil water enrichment. Another factor that the authors have not acknowledged is the potential evaporative enrichment of soil water before root absorption. Rain water can evaporate with high temperature and VPD, therefore increasing $\delta^{18}\text{O}$. Please take this into account or clarify what your assumptions are in this regard.

We agree that this requires further clarification. We have included this effect in the list of uncertainties mentioned in the results section (line 189), and have included the following

section in the supplementary discussion, to explain why we do not think this effect influences the observed $\delta^{18}O_{TR}$ trends:

“Evaporative enrichment of soil water

We assumed in our analysis that variation in tree ring $\delta^{18}O_{TR}$ largely reflected variation in rainwater $\delta^{18}O$ and not variation in soil evaporative enrichment. Soil water evaporative enrichment primarily occurs in the topsoil and is driven by the soil-to-air vapor pressure difference (Sprenger et al., 2015). Due to high vapour pressure of tropical forest understories, evaporation from the top soil is minimal and most of water vapor is derived from transpiration of plants (Moreira et al. 1997). As plant transpiration does not cause isotopic fractionation (Gat & Matsui 1991), evaporative enrichment of soil water is expected to be limited under the canopy of tropical forests. Furthermore, for soil water evaporative enrichment to exhibit temporal trends, substantial changes in the soil-to-air vapor pressure difference would be required. Given that atmospheric VPD has remained stable at our sites during the tree growing season (Supplementary Figure 8), it is unlikely that the soil-to-air vapor pressure gradient has changed significantly over time, and we thus conclude that it is unlikely that our $\delta^{18}O_{TR}$ records are influenced by changes in the degree of evaporative enrichment of soil water.”

• Supplementary L28-33 and Supplementary Table 2. I can't follow the conclusion that isotopic enrichment has a minor effect on long-term rainfall trends

. How did the authors evaluate the evaporative enrichment and stomatal conductance (g_s) responses to vapor pressure deficit? Maybe I'm a bit lost because I still don't find all the parameters in Table S2 described (view my comment on first review). E.g. What do $\delta^{18}O_{air}$, $\delta^{18}O_{sw}$, $\Delta^{18}O_{es}$, etc. mean? Please make sure you add full names and describe all the parameters you mentioned here and throughout the manuscript.

We have reviewed Table S2 and included names and descriptions of all the parameters from the equations. There was indeed one missing parameter in the descriptions ($p_x p_{ex}$), which we have now included.

We have also reviewed the related Table S1 and updated parameter values calculated for the specific seasons used in the reviewed versions of the paper (Aug-Oct for dry season and Nov-Mar for wet season).

To estimate evaporative enrichment / stomatal conductance responses to an increase of VPD we proceeded as follows:

- (i) *We calculate g_s as a function of VPD as in line last line of the second section of Table S2: “**The average $\delta^{18}O$ of leaf lamina water...**”. VPD is obtained from local climate data (CRU TS). This information is also present in the caption of Supplementary Figure 9.*
- (ii) *Both VPD and stomatal conductance affect evapotranspiration rates (E), which is an important parameter in the definition of the Péclet effect (ϕ), necessary for estimating the effect of back-diffusion of $H_2^{18}O$ from the sites of evaporation on the $\delta^{18}O$ of the leaf lamina water. The parameter ϕ is also defined in the same section of the table.*

- (iii) We then use the second main equation of the table (first line of the middle section of the table), which includes the parameter ϕ , to calculate leaf water enrichment ($\Delta^{18}\text{O}_{lw}$).
- (iv) Finally, the results from $\Delta^{18}\text{O}_{lw}$ are used in the last main equation (third section of the table) to calculate cellulose $\Delta^{18}\text{O}$ ($\Delta^{18}\text{O}_{TR}$), which takes into account post-photosynthesis fractionation effects, given by the parameters $\rho_{ex}\rho_x$.

The total calculated $\Delta^{18}\text{O}$ resulting from the direct effects of VPD and from g_s responses to VPD on leaf water enrichment from 1980 to 2010 are $0.07 \pm 0.06\text{‰}$ for the dry season chronology and $0.12 \pm 0.13\text{‰}$ for the wet season chronology, which is very small in comparison to the total $\delta^{18}\text{O}_{TR}$ trends that we observe over the same period.

2) Comments regarding the seasons used.

- L107-117: Growing seasons are still confusing.

We have made the following alterations to clarify that the periods of peak correlations are mostly within the growing seasons, but are not exact matches to them (lines 112-116, changes in bold):

*“Peak rainfall correlations for the floodplain $\delta^{18}\text{O}_{TR}$ chronology occurred from August to November, **within the** growing season for *M. acaciifolium* **which extends from July to November** (Figure 2a-c), while the terra firme $\delta^{18}\text{O}_{TR}$ chronology correlations peak from January to May, **almost entirely within** the growth season for *C. odorata* **which extends from November to April** (Figure 2d-e).”*

- Fig. 1C: What do the vertical lines represent (continuous and dotted)? This figure looks like a good opportunity to graphically represent and help clarify the different seasons you described during your manuscript:

- growing season,
- wet and dry season,
- non-flooded (terrestrial) period,
- peak wet and peak dry season used for rainfall $\delta^{18}\text{O}$ data
- growing seasons for fig 2

We've added arrows to panel (c) to better indicate the local growing seasons. We also added the following text in the figure caption to clarify:

“In (c), Arrows indicate the growing seasons of the trees, and vertical lines indicate the seasonal boundaries defined by rainfall and river levels. Vertical dotted lines extends the boundaries of the Amazon-wide seasons (bottom panel) for a visual comparison of how they overlap with the local growing seasons at each of the sites. Note that for calculations using rainfall data, we disregarded the shoulder months of those seasons, i.e. the months immediately next to the black vertical lines (details in Methods).”

- Fig S2: peak wet/dry season does not seem to coincide with the climograms of Fig 1c (dry = Aug-Oct and wet = Nov-March). Also, Fig 3 uses a different period for the dry season (i.e. = June-Oct)

That is correct, we use only the peak dry season as is described in methods lines 398-402, “we used the months from August to October for the peak dry season and from November to March for the peak wet season. These periods disregard the shoulder months from the growing season of the trees, to avoid an effect of year-to-year variability in the growing season length on our results.”

In Fig. 3, the season considered for the analyses was Aug-Oct. The y-axis was out-dated, and should have been changes as of the previous version. We have fixed this now, thank you for pointing that out.

- In the responses-to-reviewers file you mentioned that “For the terra firme trees, the growing season is defined as the period with rainfall larger than 100mm month⁻¹. For the floodplain trees the growing season is defined based on the terrestrial phase from July to November. This is graphically explained in Figure 1c.” By looking at Fig. 1c this seems more like 150 mm. Please clarify this threshold and mention it in the manuscript. Also, see my previous comment on Fig 1c.

In the second panel from Figure 1c (the terra firme panel), during all months in between the black vertical lines which define the growing season rainfall is larger than 100mm, while during all months outside that period rainfall is less than 100mm..

We have added this information to the manuscript on lines 325-326 (changes in bold):

*“C. odorata grows during the local wet season, **defined as the period of the year with rainfall exceeding 100mm** (which is October to April at this site)”*

- Why, if floodplain trees grow from July to November, you defined a different growing season in Fig. 2?

This was a mistake, thanks for pointing that out. We have now corrected the position of the vertical dotted line from Figure 2a-d.

- In the response to reviewers you added some explanation about the extended correlations beyond the growing season for C. odorata. To help clarify the confusion arising when using different groups of months during the manuscript, you could add this explanation.

We have added this explanation to lines 116-118:

“Correlations that extend outside the trees’ growing seasons are generally weaker, which could be due to interannual variation in the growing season length of the trees.”

Minor comments:

Fig S1: What does the inset plot represent?

The inset plot was there as a generic example, solely for the purpose of showing the scale of rainfall and river level of the climatograms. As the purpose of this figure is to show the timing of the river and rainfall seasons, we found that it is effectively not needed and have now removed the inset plot.

L764-766: This is a bit difficult to follow. Mixing 1-month with three-month periods. Please be consistent.

*We have reworded these lines as follows (changes in **bold**):*

*“Months January to May **in the three-month periods on the right-hand side of the x-axis correspond to the year after the onset of the ring formation.**”*

L132-134 ; 138-139 and 144-147: check consistency between text and figures.

We have checked and did not find any inconsistencies. Maybe there was an impression of inconsistencies because the $\delta^{18}O$ axis is inverted. This is done because it is negatively associated with rainfall, thus it becomes easier to interpret changes this way. We have indicated in the figure captions from Figs. 3 and 4 that the $\delta^{18}O$ axis is inverted.

L273-275: highest and lowest of which period?

We have reworded this sentence as follows (lines 281-282, changes in bold):

*“The four highest **flood-season river levels** (2021, 2012, 2009 and 2022) and three lowest **dry-season river levels** (2024, 2023, 2010) all occurred during the last 15 years⁷⁰.”*

Supplementary section “Assessment and quantification of uncertainties to estimates of rainfall changes inferred from trends in $d18OTR$ ”. Please clarify what period do you refer to when talking about rainfall predictions along these paragraphs.

We have changed this title to (changes in bold):

*“Assessment and quantification of uncertainties to estimates of rainfall changes inferred from trends in $\delta^{18}O_{TR}$ **from 1980-2010**”*

Supplementary L58-59: “only small effects on predictions in rainfall trends”. Table 1 shows up to 13% change in rainfall prediction.

Yes, that is correct. The case 0 with no change in any parameters shows a change of 13.5% (for the dry season). Altering the parameter from case 2 changes predictions to 13%. Thus, the effect of changing this parameter is 0.5%, which we refer to as being a small effect.

Table 1 and Supplementary L74-76: check the signs in the table. Also, changes in rainfall prediction seem to be much higher than 1%.

The value of 1% indicates how much case 8 differs from case 4, as stated in bold in that section of the table “Sensitivity of case 4 to”. Predictions of case 8 deviate from case 4 by approximately 1% (less so for the dry season).

We understand the confusion about assessing the effect of each parameter from the table. To clarify, we have added the following statements to the table caption:

“Several model predictions are shown, each incorporating one or more changes to the baseline parameters (Case 0). The effects of these parameter changes are reflected in the differences between Cases 1–5 and Case 0, and between Cases 6–8 and Case 4. All predictions are reported both as percentage changes per decade and as percentage change from 1980–2010, each relative to 1980 values.”

Supplementary Figure 8: Is panel “f” at the bottom of the figure actually “e”?

Yes, thank you for pointing this out. We have made this correction.

L439-440: If you refer to 1980-2010 this would be a total of 30 years

The trends themselves are analysed for longer (Figure 2), totalizing 40 years for the dry season record and 35 years for the wet season record. However, as we restricted the quantification of changes to 1980 to 2010 we agree that it makes sense to refer to 30 years here, and we have made this change.

Fig. 3B: Can you explain what does the equation on the top of the figure represent and link it with the model explained in the Methods section.

We have clarified this in the figure caption as follows (changes in bold):

*“...(b) Graphic representation of the changes in the $\delta^{18}\text{O}$ of atmospheric moisture based on a Rayleigh distillation model (**shown on top of the graph, rearranged from Eq. [1]**)...”*

To make this connection with Equation [1] clearer, we also reverted the notation ε back to $(\alpha - 1)$ in the equation from the figure.

Reviewer #3 (Remarks to the Author):

I applauded that the authors made additional analyses and interpretation to address my concerns. I recommend the manuscript for publication after below edit is done.

L45: Local-scale deforestation increases precipitation, whereas large-scale deforestation reduces precipitation (Spracklen et al., 2018). So it is "Large-scale

deforestation could cause reductions in precipitation...", not "...cause large-scale precipitation"

We have made this change.